# A discrete parasubthalamic nucleus subpopulation plays a critical role in appetite suppression

Jessica H Kim[1†], Grace H Kromm[1†], Olivia K Barnhill[1†], Jacob Sperber[1], Lauren B Heuer[1], Sierra Loomis[1], Matthew C Newman[1], Kenneth Han[1], Faris F Gulamali[1], Theresa B Legan[1], Katharine E Jensen[2], Samuel C Funderburk[3], Michael J Krashes[3], Matthew E Carter[1]*

[1]Department of Biology, Williams College, Williamstown, United States; [2]Department of Physics, Williams College, Williamstown, United States; [3]Diabetes, Endocrinology, and Obesity Branch, National Institute of Diabetes and Digestive and Kidney Diseases, Bethesda, United States

*For correspondence:
mc10@williams.edu

[†]These authors contributed equally to this work

Competing interest: The authors declare that no competing interests exist.

**Abstract** Food intake behavior is regulated by a network of appetite-inducing and appetite-suppressing neuronal populations throughout the brain. The parasubthalamic nucleus (PSTN), a relatively unexplored population of neurons in the posterior hypothalamus, has been hypothesized to regulate appetite due to its connectivity with other anorexigenic neuronal populations and because these neurons express Fos, a marker of neuronal activation, following a meal. However, the individual cell types that make up the PSTN are not well characterized, nor are their functional roles in food intake behavior. Here, we identify and distinguish between two discrete PSTN subpopulations, those that express tachykinin-1 (PSTN$^{Tac1}$ neurons) and those that express corticotropin-releasing hormone (PSTN$^{CRH}$ neurons), and use a panel of genetically encoded tools in mice to show that PSTN$^{Tac1}$ neurons play an important role in appetite suppression. Both subpopulations increase activity following a meal and in response to administration of the anorexigenic hormones amylin, cholecystokinin (CCK), and peptide YY (PYY). Interestingly, chemogenetic inhibition of PSTN$^{Tac1}$, but not PSTN$^{CRH}$ neurons, reduces the appetite-suppressing effects of these hormones. Consistently, optogenetic and chemogenetic stimulation of PSTN$^{Tac1}$ neurons, but not PSTN$^{CRH}$ neurons, reduces food intake in hungry mice. PSTN$^{Tac1}$ and PSTN$^{CRH}$ neurons project to distinct downstream brain regions, and stimulation of PSTN$^{Tac1}$ projections to individual anorexigenic populations reduces food consumption. Taken together, these results reveal the functional properties and projection patterns of distinct PSTN cell types and demonstrate an anorexigenic role for PSTN$^{Tac1}$ neurons in the hormonal and central regulation of appetite.

## Editor's evaluation

This work has identified a previously unrecognized role of the parasubthalamic nucleus in the regulation of feeding behavior. The combination of genetic and pharmacological approaches nicely demonstrates the physiological role of this group of neurons in regulating appetite. These studies will be of interest to the field and more broadly to the readers of *eLife*.

## Introduction

The brain regulates food intake behavior through the coordinated activity of several distinct neuronal populations (*Andermann and Lowell, 2017*; *Sternson and Eiselt, 2017*). Activity in orexigenic

(appetite-inducing) populations, such as agouti-related peptide (AgRP)-expressing neurons in the hypothalamic arcuate nucleus, can rapidly and reversibly promote an increase in food intake (*Aponte et al., 2011*; *Krashes et al., 2011*). In contrast, activity in anorexigenic (appetite-suppressing) populations, such as pro-opiomelanocortin (POMC)-expressing neurons in the hypothalamus (*Aponte et al., 2011*) or calcitonin gene related peptide (CGRP)-expressing neurons in the brainstem parabrachial nucleus (PBN[CGRP] neurons) (*Campos et al., 2016*; *Carter et al., 2013*), suppresses feeding. In recent years, the parasubthalamic nucleus (PSTN), a relatively understudied population of neurons in the posterolateral hypothalamus, has been hypothesized to regulate feeding behavior (*Shah et al., 2022*); however, the gene expression patterns within the PSTN and the roles of individual cell types in food intake behavior have remained relatively uncharacterized.

The few studies that have explored the PSTN demonstrate a potential role for these neurons in feeding. Anatomically, PSTN neurons receive afferent projections from orexigenic AgRP neurons (*Livneh et al., 2017*; *Wang et al., 2015*) and anorexigenic populations including arcuate POMC neurons (*Wang et al., 2015*), PBN[CGRP] neurons (*Huang et al., 2021*), and the central nucleus of the amygdala (CeA) (*Barbier et al., 2017*). In turn, these neurons project to anorexigenic brain regions including the PBN (*Goto and Swanson, 2004*; *Zséli et al., 2016*), CeA (*Barbier et al., 2020*; *Zséli et al., 2018*), nucleus of the solitary tract (NST) (*Ciriello et al., 2008*; *Goto and Swanson, 2004*; *Holt et al., 2019*), and paraventricular thalamic nucleus (PVT) (*Zhang and van den Pol, 2017*). PSTN neurons express Fos, an immediate early gene that serves as an indirect marker of neuronal activation, following a large meal (*Barbier et al., 2020*; *Chometton et al., 2016*; *Zséli et al., 2016*; *Zséli et al., 2018*), during a learned conditioned flavor aversion (*Yasoshima et al., 2006*), and in response to dietary amino acid deficiency that causes a decrease in appetite (*Zhu et al., 2012*). Optogenetic stimulation of glutamatergic projections from the PSTN to the PVT reduces food intake (*Zhang and van den Pol, 2017*). Finally, chemogenetic inhibition of PSTN neurons attenuates the appetite suppressing effects of the hormone cholecystokinin (*Sanchez et al., 2022*), and chemogenetic inhibition of PSTN neurons that express tachykinin-1 (Tac1) decreases taste neophobia (*Barbier et al., 2020*).

Taken together, these initial studies suggest a role for the PSTN in feeding behavior. However, the individual subpopulations that make up the PSTN remain relatively uncharacterized, as do their functional roles during food consumption. We therefore sought to characterize the cell types that make up the PSTN, determine their activity patterns in response to various appetitive stimuli, perturb their function in freely behaving mice, and map their connectivity with other brain regions. We identify two discrete populations of PSTN neurons, each with distinct influences on feeding behavior.

## Results

### Identification of distinct PSTN cell populations

We initially identified the PSTN as a candidate anorexigenic population by investigating sources of afferent input to PBN[CGRP] neurons, a population that has previously been shown to suppress appetite and control meal termination (*Campos et al., 2017*; *Campos et al., 2016*; *Carter et al., 2013*; *Essner et al., 2017*). To retrogradely label neurons that project to PBN[CGRP] neurons, we used a Cre-inducible modified rabies virus system in *Calca*[Cre/+] mice (*Calca* is the gene that encodes CGRP; *Figure 1A and B*). Retrograde expression was observed in populations previously shown to project to the PBN, including the bed nucleus of the stria terminalis (BNST) (*Wang et al., 2019*), central nucleus of the amygdala (CeA) (*Cai et al., 2014*), arcuate nucleus of the hypothalamus (Arc) (*Carter et al., 2013*; *Essner et al., 2017*; *Wu et al., 2009*), and nucleus of the solitary tract (NTS) (*Roman et al., 2016*; *Figure 1C–H*). We also observed retrogradely labeled cells in the PSTN, just medial to the cerebral peduncle in the posterior hypothalamus (*Figure 1G and I*). Similar to previous studies (*Barbier et al., 2020*; *Chometton et al., 2016*; *Zséli et al., 2016*; *Zséli et al., 2018*), we found that refeeding following 18 hr food deprivation induced substantial expression of Fos in the PSTN region (*Figure 1J and K*), suggesting that PSTN neurons play a role in appetite suppression. Therefore, to better characterize these Fos-expressing neurons, we analyzed the cell types that make up the PSTN and explored their roles in food intake behavior.

To identify potential genetic markers for PSTN neurons, we consulted the Allen Brain Explorer (http://musebrain-map.org) (*Ng et al., 2009*) and searched for genes enriched in the PSTN. Additionally, previous studies reported expression of *Tac1* (*Barbier et al., 2020*; *Wallén-Mackenzie et al.,*

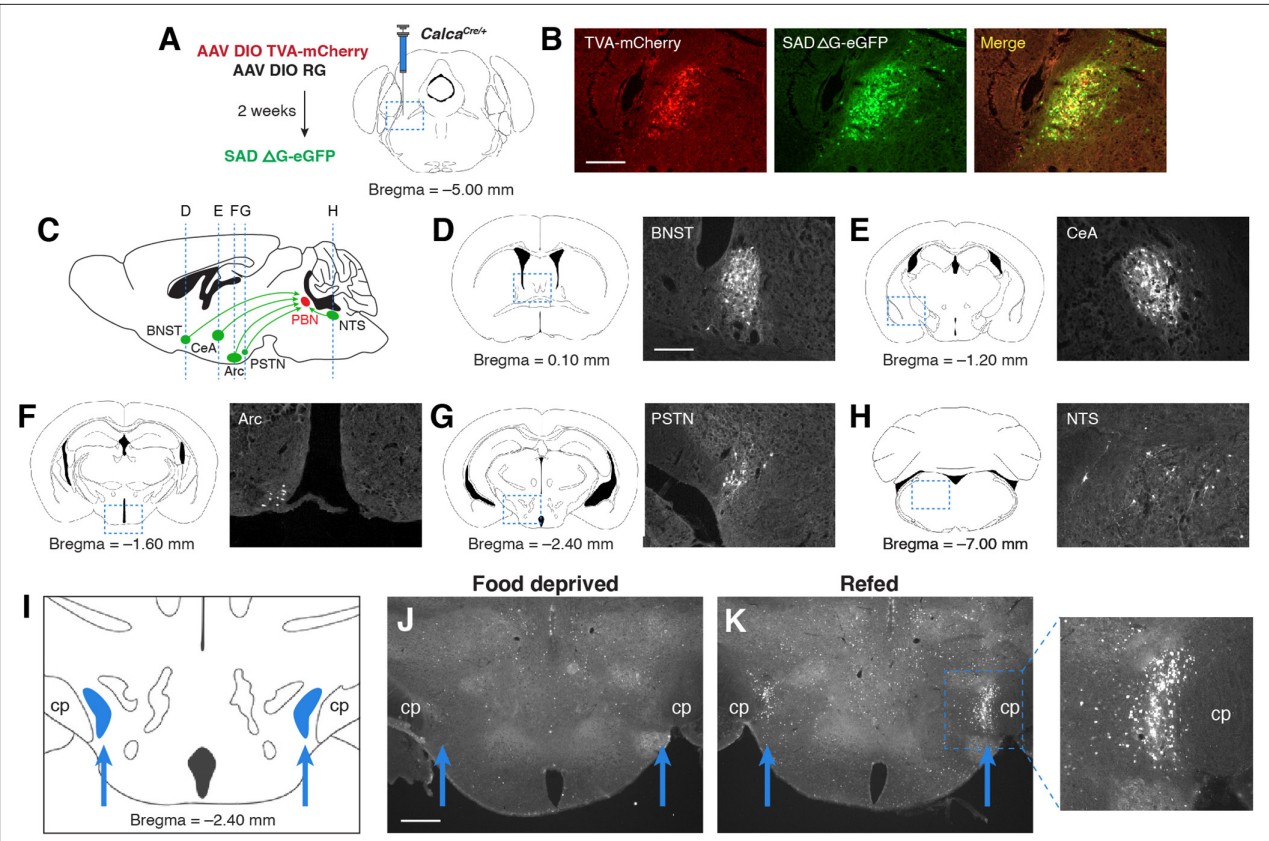

**Figure 1.** PSTN neurons project to anorexigenic PBN[CGRP] neurons and are activated by refeeding. (**A**) Modified rabies viral strategy used to identify afferent input to PBN[CGRP] neurons. (**B**) Confirmation of modified rabies virus reporter transgenes in PBN[CGRP] neurons. Experiments repeated in N=6 mice. Scale bar, 250 µm. (**C**) Sagittal mouse brain diagram showing locations of retrograde eGFP expression in (**D–H**). (**D–H**) eGFP-expressing neurons in (**D**) the bed nucleus of the stria terminalis (BNST); (**E**) the central nucleus of the amygdala (CeA); (**F**) the hypothalamic arcuate nucleus (Arc); (**G**) the parasubthalamic nucleus (PSTN); and (**H**) the nucleus of the solitary tract (NTS). Scale bar, 250 µm. (**I**) Coronal mouse brain diagram depicting the bilateral location of the PSTN medial to the cerebral peduncle (cp). Blue arrows point to the location of the PSTN. (**J**) Immunolabelling of Fos expression following 18 hr food deprivation. Scale bar, 500 µm. (**K**) Immunolabelling of Fos expression following 18 hr food deprivation followed by 30 min refeeding. Right, higher magnification image.

*2020*) and *Crh* (*Zhu et al., 2012*) in the PSTN region. Indeed, the Allen Mouse Brain Atlas (*Lein et al., 2007*) demonstrates expression of *Tac1* and *Crh* within the PSTN region, with expression of *Slc17a6* (the gene that encodes VGlut2), *Calb1*, *Calb2*, and *Pvalb* more diffusely located throughout the local area (*Figure 2A*). Expression of the GABAergic markers *Gad1* and *Gad2* are notably absent from the PSTN region (*Shah et al., 2022*).

To quantify the specificity and co-expression of genetic markers in the PSTN, we performed fluorescent in situ hybridization (FISH), initially focusing on *Tac1* and *Crh* due to the apparent specificity of these markers within the PSTN. As expected, *Tac1*- and *Crh*-expressing cells were located within the PSTN region (*Figure 2B*). Interestingly, there was nearly distinct expression of each marker in the PSTN, with 85.1% of labeled PSTN cells expressing *Tac1*, 14.9% of labeled PSTN cells expressing *Crh*, and only 1.9% of cells co-expressing both markers (*Figure 2C*). Therefore, PSTN[Tac1] and PSTN[CRH] neurons constitute two nearly distinct cell populations in the PSTN.

To determine the co-expression of PSTN[Tac1] and PSTN[CRH] neurons with other potential genetic markers, we performed triple-label FISH experiments. Every *Tac1*- and *Crh*-expressing cell also expressed *Slc17a6*, indicating that these populations are glutamatergic (*Figure 2D and E*). Although it is impossible to delineate a precise border for the PSTN using *Slc17a6* because adjacent regions are also glutamatergic, we estimate that ~22% of *Slc17a6*-expressing neurons within the PSTN region do not express either *Tac1* or *Crh*, indicating the presence of glutamatergic PSTN cell types that may express other unique genetic markers. A majority of *Tac1*- and *Crh*-expressing cells co-expressed

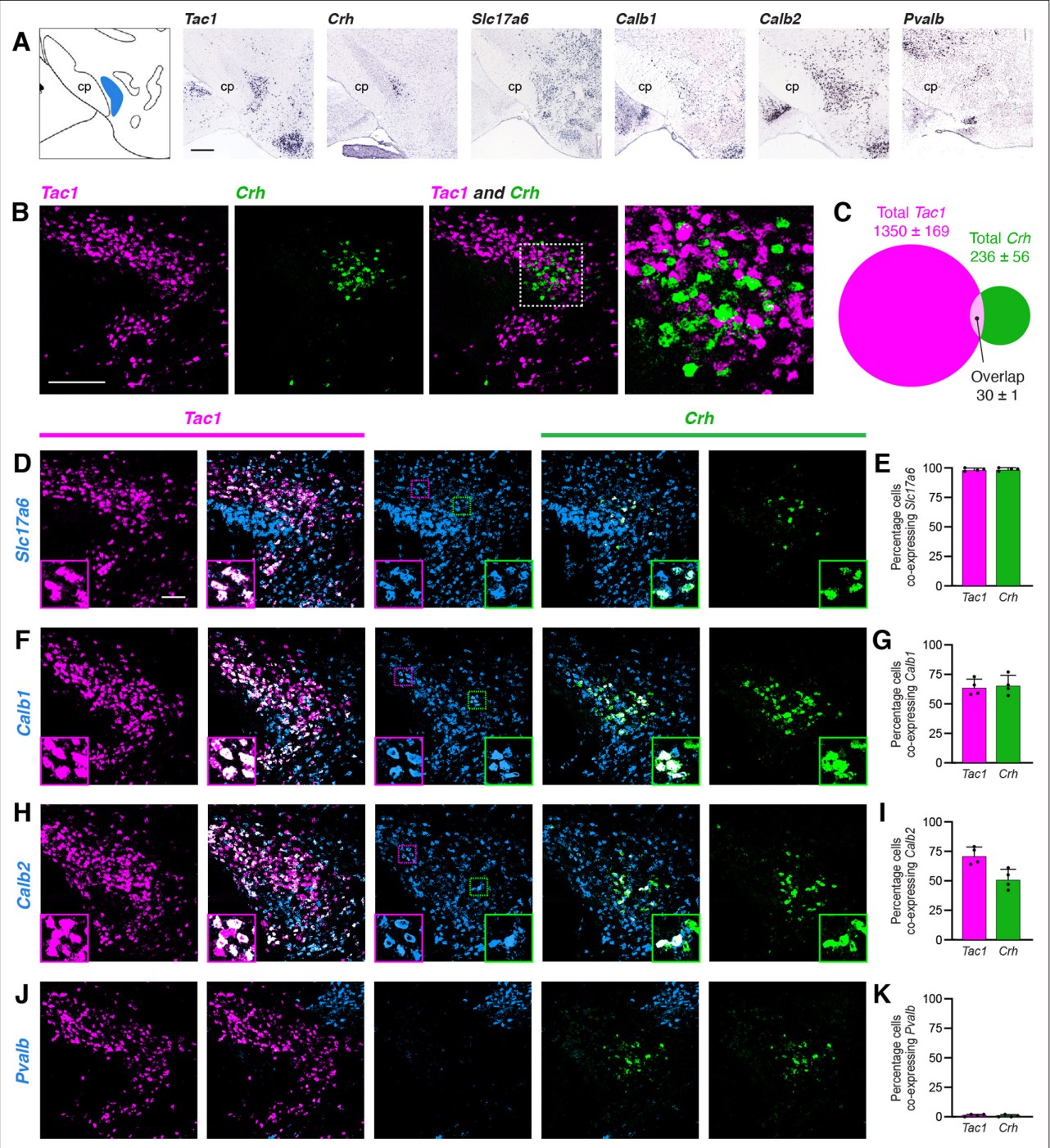

**Figure 2.** Characterization of PSTN cell types. (**A**) In situ hybridization images of selected mRNA transcripts from the Allen Mouse Brain Expression Atlas. cp, cerebral peduncle. Scale bar, 500 µm. (**B**) Two-color fluorescent in situ hybridization shows near distinct expression of *Tac1* and *Crh* in PSTN neurons. Right, higher magnification image. Scale bar, 200 µm. (**C**) Quantification of overlap between PSTN[Tac1] and PSTN[CRH] neurons. N=4 mice. (**D–K**) Three-color fluorescent in situ hybridization comparing spatial overlap between neurons expressing *Tac1* and *Crh* with (**D,E**) *Slc17a6*; (**F,G**) *Calb1*; (**H,I**) *Calb2*; and (**J,K**) *Pvalb*. Insets show higher magnification images from middle column. Data at right represent mean ± standard deviation. Dots represent individual experimental animals; N=4 mice. Scale bar, 100 µm.

The online version of this article includes the following source data for figure 2:

**Source data 1.** This file contains the raw data for the visualization of data presented in *Figure 2*.

*Calb1* and *Calb2*, although expression of these markers was diffuse and extended medially far beyond the PSTN region (*Figure 2F–I*). Less than 2% of *Tac1* and *Crh*-expressing cells co-expressed *Pvalb* (*Figure 2J and K*). A *Pvalb*-expressing population appeared dorsomedial to the PSTN, while two other *Pvalb* populations appeared medially and laterally to the PSTN in the same horizontal plane. Therefore, Tac1 and CRH neurons are specific and discrete glutamatergic subpopulations within the PSTN.

## PSTN neurons increase activity in response to feeding and anorexigenic hormones

To determine whether PSTN^Tac1 and/or PSTN^CRH neurons are active in response to feeding, we first pursued a histological approach using in situ hybridization to measure co-expression of *Fos* following a large meal. Mice were food deprived for 18 hr; half the animals were then allowed to consume a large meal while the other half remained fasted, and all animals were perfused 45 min later to allow time for *Fos* mRNA expression dynamics. Animals that were not allowed to eat following the food deprivation period exhibited relatively low *Fos* expression in the PSTN (*Figure 3A*). In contrast, animals that were allowed to refeed showed robust co-expression of *Fos* in both *Tac1* and *Crh* populations (*Figure 3B and C*; See also *Supplementary file 1* for detailed statistical analysis). Although a greater percentage of *Crh* neurons co-expressed *Fos*, a significantly higher total number of *Tac1* neurons co-expressed *Fos* (*Figure 3D*). Taken together, these results demonstrate that refeeding increases activity as measured by *Fos* expression in both PSTN^Tac1 and PSTN^CRH neurons.

To determine acute activity patterns in PSTN^Tac1 or PSTN^CRH neurons upon food exposure, we transduced PSTN neurons in *Tac1^Cre/+* or *Crh^Cre/+* mice with the GCaMP6s calcium indicator and measured real-time fluorescence intensity using fiber photometry (*Figure 3E and F*). Although we detected post-hoc GCaMP6s fluorescent expression in *Crh^Cre/+* mice, we were not able to record reliable fluorescent signals from these mice following a variety of conditions, potentially due to the relatively low number of CRH neurons in the PSTN. Therefore, we focused on *Tac1^Cre/+* mice, recording from mice either food deprived for 18 hr or fed ad libitum. Exposure to standard mouse chow or palatable peanut butter produced a significant increase in PSTN^Tac1 activity in 18 hr food deprived mice compared with exposure to a novel object or exposure to water in 16 hr water deprived mice (*Figure 3G–L*). This increase in activity persisted for approximately the first 60 s upon exposure to food. These effects were diminished in ad libitum fed mice that were less motivated to consume food. Taken together, these results suggest that PSTN^Tac1 neurons increase activity during the initial stages of food consumption in hungry mice, but do not respond to water in thirsty mice or other salient stimuli.

We next examined the effects of anorexigenic hormone administration on activity in PSTN^Tac1 and PSTN^CRH neurons. To measure neural activity using *Fos* expression, animals were food deprived for 18 hr to reduce endogenous anorexigenic hormone levels, and then injected intraperitoneally with saline, amylin, cholecystokinin (CCK), or peptide YY (PYY). Administration of these anorexigenic hormones caused an increase in *Fos* expression in both *Tac1* and *Crh* neurons compared with animals injected with saline (*Figure 4A–E*), indicating that these anorexigenic hormones cause increased activation of PSTN^Tac1 and PSTN^CRH neurons. To measure neural activity using fiber photometry, *Tac1^Cre/+* mice transduced with GCaMP6s were implanted with an intraperitoneal catheter (*Figure 4F*), allowing for real-time measurement of PSTN^Tac1 activity during hormone administration without the need to inject mice and potentially cause stress by physically handling animals. Administration of amylin, CCK, and PYY caused a significant increase in PSTN^Tac1 activity compared with administration of saline (*Figure 4G–L*). Therefore, PSTN^Tac1 neurons increase activity during food consumption and in response to elevated levels of anorexigenic hormones.

## Inhibition of PSTN^Tac1 neurons attenuates the effects of anorexigenic hormones

To determine whether activity in PSTN^Tac1 or PSTN^CRH neurons is necessary for normal food intake behavior, we bilaterally injected AAV carrying either Cre-inducible hM4Di-mCherry or mCherry transgenes into the PSTN of *Tac1^Cre/+* and *Crh^Cre/+* mice (*Figure 5A–C*).

In *Tac1^Cre/+* mice, there was no difference in food consumption between animals transduced with hM4Di-mCherry and mCherry following intraperitoneal administration of clozapine *N*-oxide (CNO) and inert saline solution (*Figure 5D*). Therefore, inhibition of PSTN^Tac1 neurons is insufficient to affect

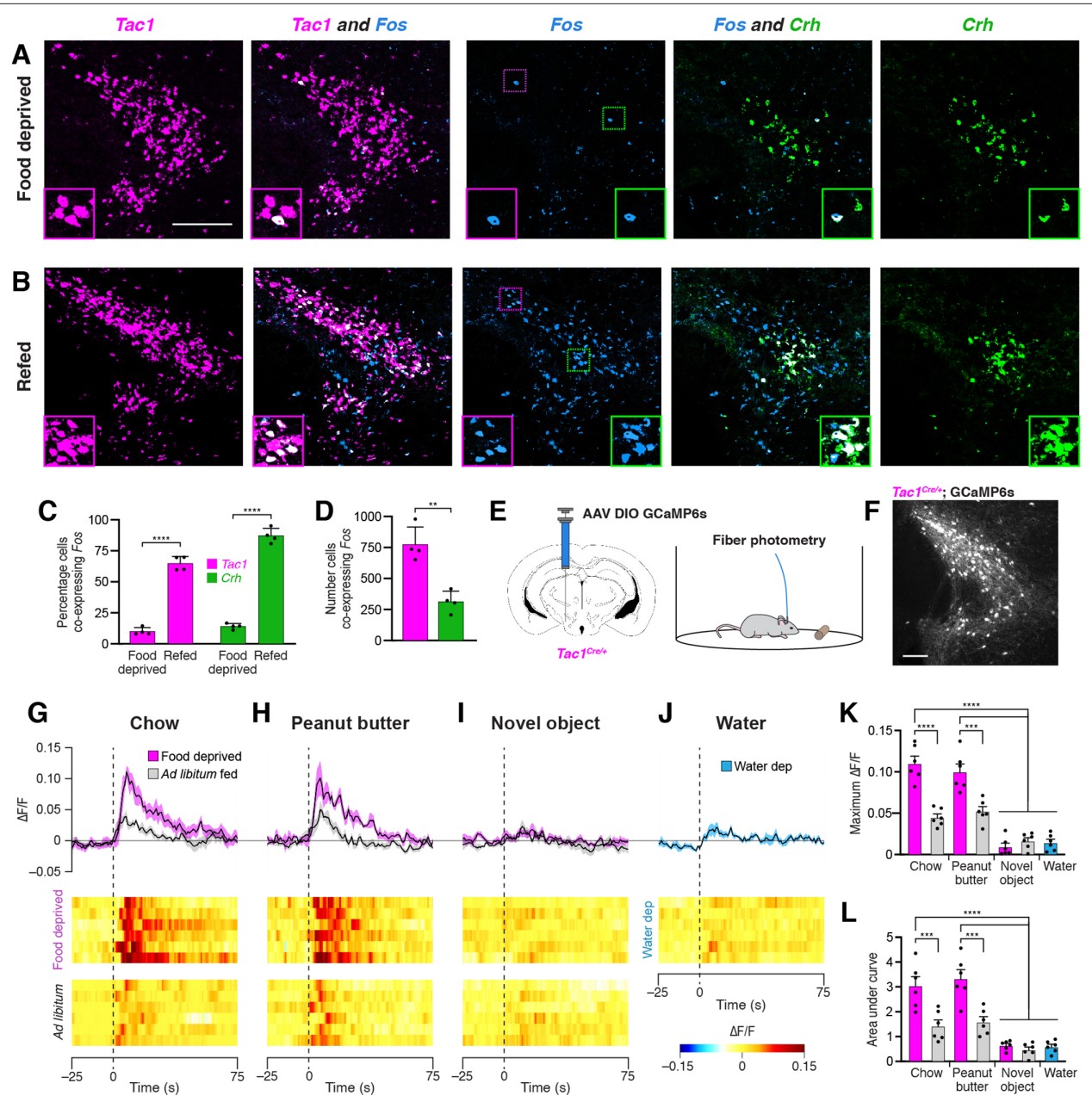

**Figure 3.** Refeeding and food consumption activate PSTN neurons. (**A–B**) Three-color fluorescent in situ hybridization comparing co-expression of *Fos* in neurons expressing *Tac1* or *Crh* following (**A**) 18 hr food deprivation or (**B**) 18 hr food deprivation followed by 30 min refeeding. Insets show higher magnification images from middle column. Scale bar, 200 μm. (**C**) Quantification of the percentage of *Tac1*- or *Crh*-expressing neurons co-expressing *Fos* in 18 hr food deprivation versus refeeding conditions. N=4 mice. (**D**) Quantification of the number of *Tac1*- or *Crh*-expressing neurons co-expressing *Fos* following refeeding. N=4 mice. (**E**) Diagrams showing (left) viral injection strategy to unilaterally target PSTN^Tac1 neurons with GCaMP6s and (right) subsequent fiber photometry recording in freely moving mice. (**F**) Representative photomicrograph showing PSTN^Tac1 neurons expressing GCaMP6s. Scale bar, 100 μm. (**G–J**) Top, fiber photometry traces in PSTN^Tac1 neurons following exposure to (**G**) standard mouse chow, (**H**) peanut butter, or (**I**) a novel object from animals food deprived for 18 hr (magenta) or fed ad libitum (grey). (**J**) Fiber photometry trace in PSTN^Tac1 neurons following exposure to a water port in water-deprived mice. Data represent the mean ± standard error. Vertical dashed lines depict time of exposure. Bottom, heat maps depicting changes in fluorescence intensity in individual animals. (**K**) Quantification of maximum values of fluorescence intensity in conditions (**G–J**). N=6 mice. (**L**) Quantification of area under the curve of fluorescence intensity among conditions in (**G–J**). N=6 mice. Data represent mean ± standard error. Dots represent individual experimental animals. Post hoc comparisons: **p<0.01, ***p<0.001, ****p<0.0001. See *Supplementary file 1* for additional statistical information.

The online version of this article includes the following source data for figure 3:

**Source data 1.** This file contains the raw data for the visualization of data presented in *Figure 3*.

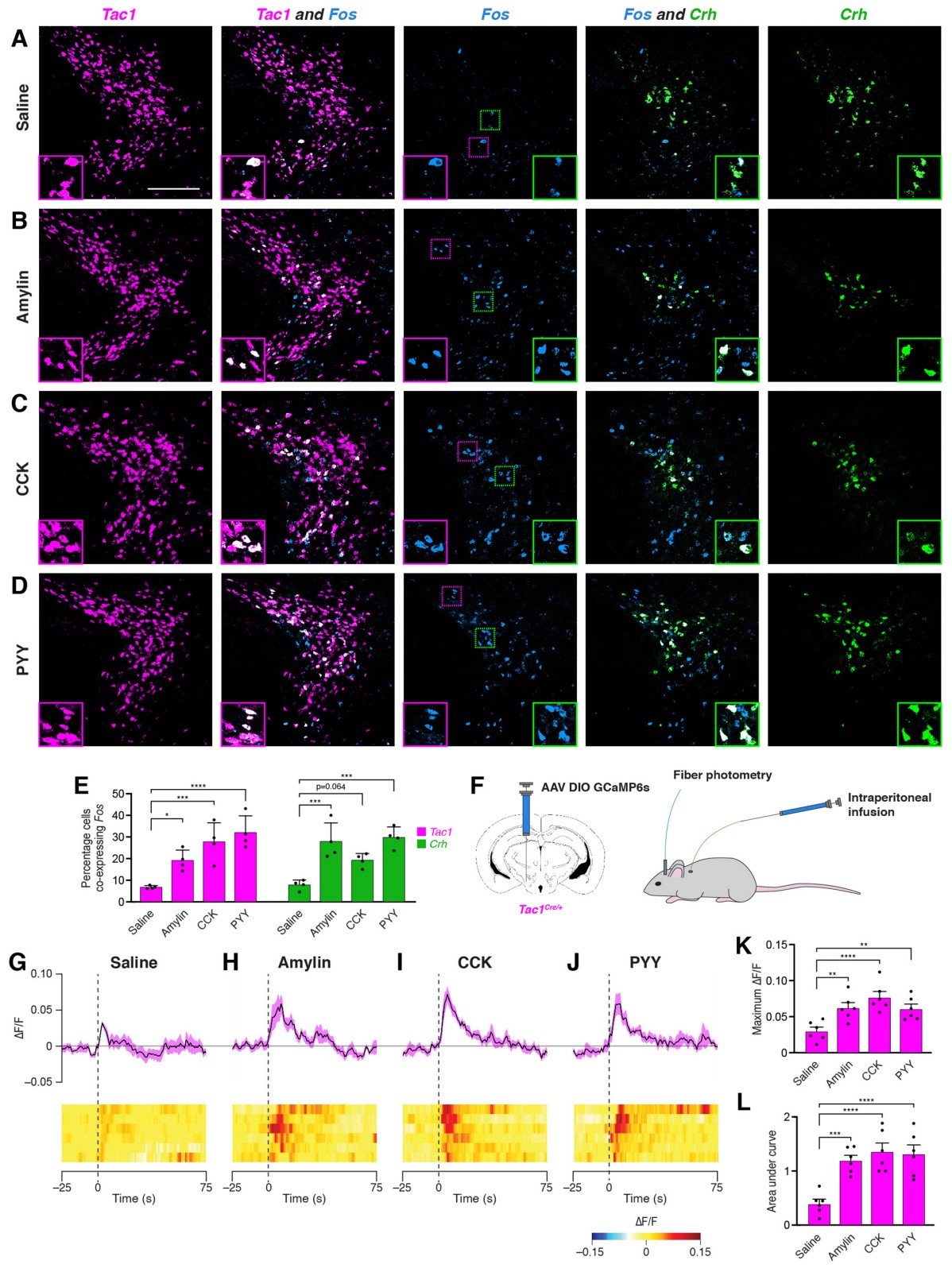

**Figure 4.** Anorexigenic hormones cause transient activation of PSTN neurons. (**A–D**) Three-color fluorescent in situ hybridization comparing co-expression of *Fos* in neurons expressing *Tac1* or *Crh* following administration of (**A**) saline, (**B**) amylin, (**C**) CCK, or (**D**) PYY. Insets show higher magnification images from middle column. Scale bar, 200 μm. (**E**) Quantification of the percentage of *Tac1*- or *Crh*-expressing neurons co-expressing *Fos* across conditions in (**A–D**). N=4 mice. (**F**) Diagram showing (left) viral injection strategy to unilaterally target PSTN^Tac1 neurons with GCaMP6s

*Figure 4 continued on next page*

*Figure 4 continued*

and (right) subsequent fiber photometry recording in freely moving mice coupled with intraperitoneal infusion of anorexigenic hormones. (**G–J**) Top, fiber photometry traces in PSTN[Tac1] neurons following intraperitoneal infusion of (**G**) saline, (**H**) amylin, (**I**) CCK, or (**J**) PYY. Data represent the mean ± standard error. Vertical dashed lines depict time of injection. Bottom, heat maps depicting changes in fluorescence intensity in individual animals. (**K**) Quantification of maximum values of fluorescence intensity in conditions (**G–J**). N=6 mice. (**L**) Quantification of area under the curve of fluorescence intensity among conditions in (**G–J**). N=6 mice. Data represent mean ± standard error. Dots represent individual experimental animals. Post hoc comparisons: *p<0.05, **p<0.01, ***p<0.001, ****p<0.0001. See *Supplementary file 1* for additional statistical information.

The online version of this article includes the following source data for figure 4:

**Source data 1.** This file contains the raw data for the visualization of data presented in *Figure 4*.

baseline food intake. Because anorexigenic hormones increased activity in PSTN[Tac1] neurons (*Figure 4*), we assessed whether this activity is necessary for their anorexigenic effects. As expected, administration of amylin, CCK, and PYY significantly reduced food intake relative to administration of saline (*Figure 5D–G*). Intriguingly, co-administration of CNO significantly attenuated these anorexigenic

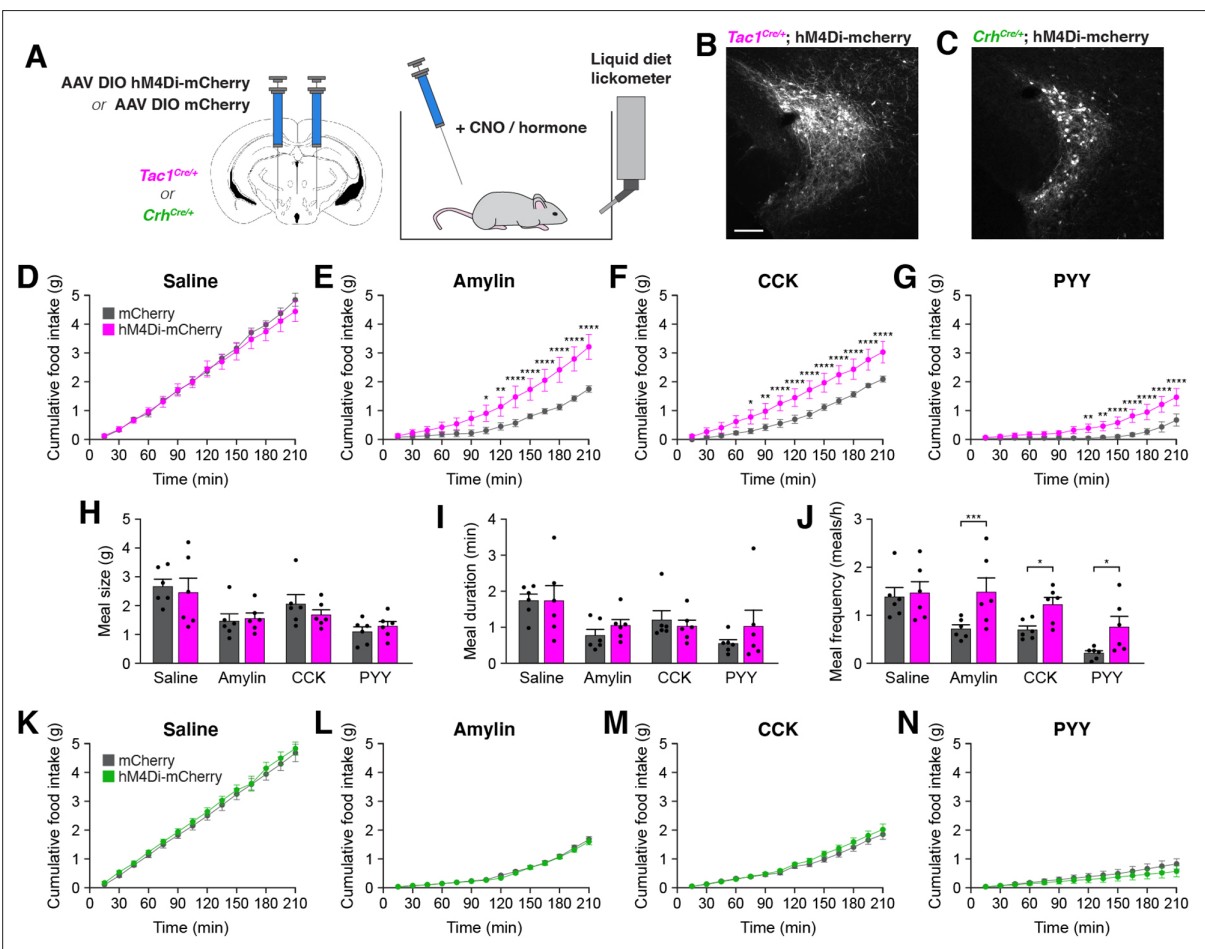

**Figure 5.** Inhibition of PSTN[Tac1] neurons attenuates the anorexigenic effects of appetite-suppressing hormones. (**A**) Diagram showing (left) viral injection strategy to unilaterally target PSTN[Tac1] or PSTN[CRH] neurons with hM4Di-mCherry or mCherry transgenes and (right) subsequent food intake measurements following administration of CNO and anorexigenic hormones. (**B**) Representative photomicrograph showing PSTN[Tac1] neurons expressing hM4Di-mCherry. Scale bar, 100 μm. (**C**) Representative photomicrograph showing PSTN[CRH] neurons expressing hM4Di-mCherry. (**D–G**) Cumulative food intake in *Tac1Cre/+* animals administered CNO followed with (**D**) saline, (**E**) amylin, (**F**) CCK, or (**G**) PYY. (**H–J**) Quantification of (**H**) meal size, (**I**) meal duration, (**J**) and meal frequency in conditions (**D–G**). (**K–N**) Cumulative food intake in *CrhCre/+* animals administered CNO followed with (**K**) saline, (**L**) amylin, (**M**) CCK, or (**N**) PYY. Data represent mean ± standard error. Dots represent individual experimental animals; N=6 mice in all experiments. Post hoc comparisons: *p<0.05, **p<0.01, ***p<0.001, ****p<0.0001. See *Supplementary file 1* for additional statistical information.

The online version of this article includes the following source data for figure 5:

**Source data 1.** This file contains the raw data for the visualization of data presented in *Figure 5*.

effects in hM4Di-transduced animals (***Figure 5E–G***), demonstrating the necessity of PSTN[Tac1] neuronal activity for the full anorexigenic effects of amylin, CCK, and PYY. To better understand how PSTN[Tac1] neuron inhibition increases food intake in these conditions, we analyzed individual meal parameters in the first 3 hr following injection of CNO. Administration of CNO did not affect meal size or meal duration, but increased meal frequency in hM4Di-mCherry-transduced mice (***Figure 5H–J***). This analysis suggests that inhibition of PSTN[Tac1] neurons attenuates the reduction in food intake caused by anorexigenic hormones by increasing the frequency of meals. In contrast to *Tac1^Cre/+* mice, there were no differences between hM4Di-mCherry and mCherry-transduced *Crh^Cre/+* mice following administration of CNO with saline (***Figure 5K***) or anorexigenic compounds (***Figure 5L–N***). Therefore, activity in

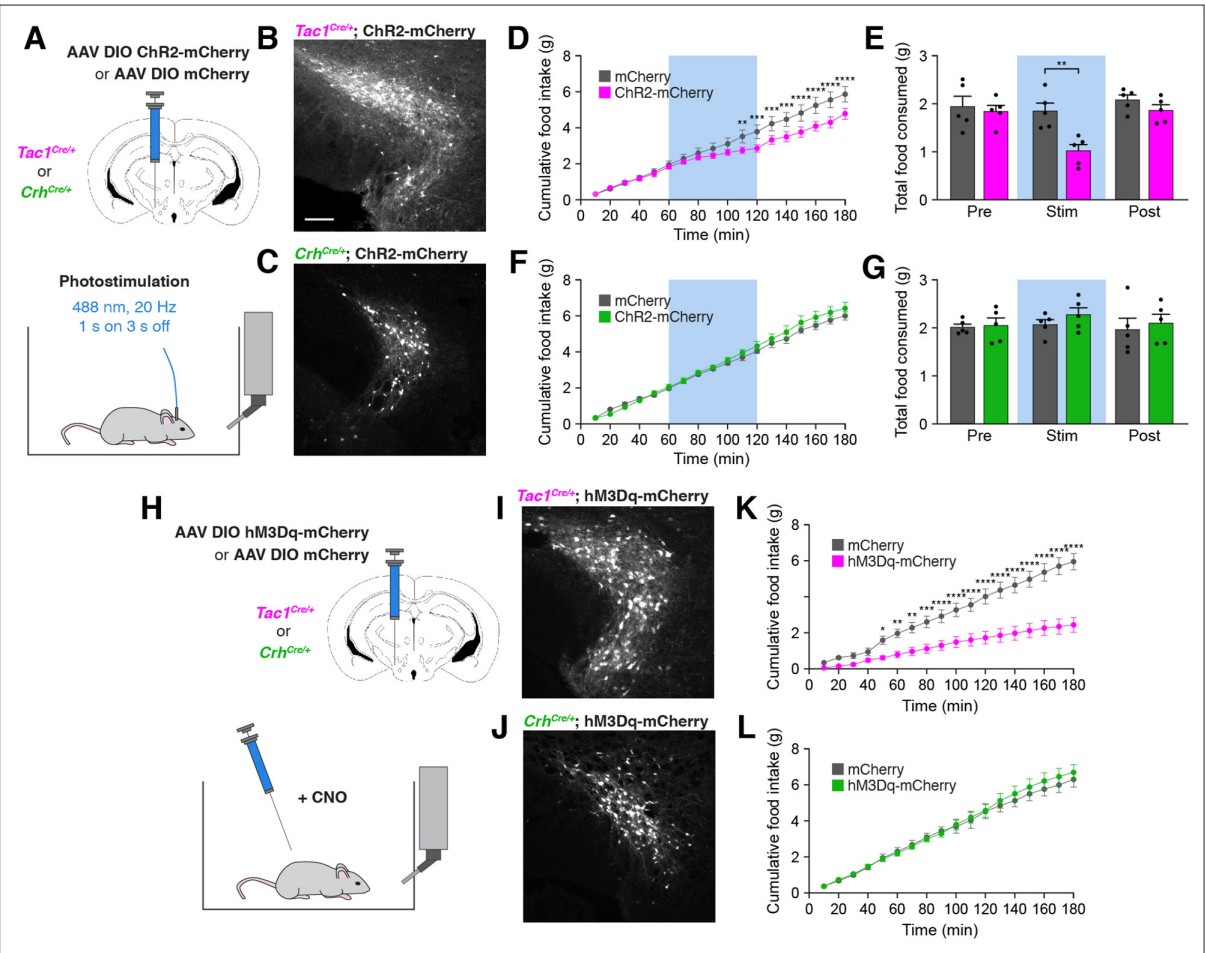

**Figure 6.** Stimulation of PSTN[Tac1] neurons suppresses feeding. (**A**) Diagram showing (top) viral injection strategy to unilaterally target PSTN[Tac1] or PSTN[CRH] neurons with ChR2-mCherry or mCherry transgenes and (bottom) subsequent optogenetic manipulation during food intake recordings. (**B**) Representative photomicrograph showing PSTN[Tac1] neurons expressing ChR2-mCherry. Scale bar, 100 μm. (**C**) Representative photomicrograph showing PSTN[CRH] neurons expressing ChR2-mCherry. (**D–E**) Quantification of (**D**) cumulative food intake and (**E**) total food consumed during pre-stimulation, stimulation, and post-stimulation of PSTN[Tac1] neurons. Blue background represents the 60-min photostimulation period. (**F–G**) Quantification of (**F**) cumulative food intake and (**G**) total food consumed during pre-stimulation, stimulation, and post-stimulation of PSTN[CRH] neurons. Blue background represents the 60-min photostimulation period. (**H**) Diagram showing (top) viral injection strategy to unilaterally target PSTN[Tac1] or PSTN[CRH] neurons with hM3Dq-mCherry or mCherry transgenes and (bottom) subsequent chemogenetic manipulation during food intake recordings. (**I**) Representative photomicrograph showing PSTN[Tac1] neurons expressing hM3Dq-mCherry. (**J**) Representative photomicrograph showing PSTN[CRH] neurons expressing hM3Dq-mCherry. (**K–L**) Quantification of cumulative food intake in (**K**) *Tac1^Cre/+* or (**L**) *Crh^Cre/+* mice following administration of CNO. Data represent mean ± standard error. Dots represent individual experimental animals; N=5 mice in all experiments. Post hoc comparisons: *p<0.05, **p<0.01, ***p<0.001, ****p<0.0001. See ***Supplementary file 1*** for additional statistical information.

The online version of this article includes the following source data for figure 6:

**Source data 1.** This file contains the raw data for the visualization of data presented in ***Figure 6***.

PSTN^Tac1 neurons, but not PSTN^CRH neurons, is necessary for the suppression of meal frequency caused by amylin, CCK, and PYY.

## Activation of PSTN^Tac1 neurons suppresses food intake

To determine whether a gain of activity in PSTN^Tac1 or PSTN^CRH neurons is sufficient to suppress feeding, we unilaterally injected AAV carrying either Cre-inducible mCherry or ChR2-mCherry transgenes into the PSTN of *Tac1^Cre/+* and *Crh^Cre/+* mice for optogenetic stimulation experiments (*Figure 6A–C*). Mice transduced with ChR2-mCherry in PSTN^Tac1 neurons consumed significantly less food during the photostimulation period than mice transduced with mCherry (*Figure 6D and E*). In contrast, photostimulation of PSTN^CRH neurons caused no observable effects on food intake (*Figure 6F and G*). To independently verify these effects and assess the impact of longer term stimulation, we also examined the effects of chemogenetic stimulation of PSTN^Tac1 or PSTN^CRH neurons on food intake behavior by transducing these neurons with mCherry or hM3Dq-mCherry transgenes (*Figure 6H–J*). Following administration of CNO, mice transduced with hM3Dq-mCherry in PSTN^Tac1 neurons consumed significantly less food over a 3 hr period than mice transduced with mCherry (*Figure 6K*). There was no effect of CNO administration between mice transduced with hM3Dq-mCherry or mCherry in PSTN^CRH neurons (*Figure 6L*). Therefore, these gain-of-function experiments suggest that optogenetic or chemogenetic stimulation of PSTN^Tac1 neurons, but not PSTN^CRH neurons, reduces food consumption.

## PSTN^Tac1 and PSTN^CRH neurons exhibit different projection patterns

The necessity and sufficiency of PSTN^Tac1 neurons, but not PSTN^CRH neurons, in appetite suppression suggests that these neural populations have different projection targets throughout the brain. To identify and distinguish between downstream projections of PSTN^Tac1 and PSTN^CRH neurons, we transduced each population with mCherry and surveyed the brain for anterograde fluorescence (*Figure 7*). PSTN^Tac1 neurons uniquely projected to the PVT, PBN, NTS, and intermediate reticular nucleus (IRT), while PSTN^CRH neurons uniquely projected to the lateral reticular nucleus (LRN). Both populations projected to the BNST, CeA, medial reticular nucleus (MRN), and tegmental reticular nucleus (TRN).

Because stimulation of PSTN^Tac1 neurons decreased feeding, we tested the effects of stimulating their projections to neuronal populations known to regulate feeding behavior. Optogenetic stimulation of projections from PSTN^Tac1 neurons to the BNST did not cause changes in food intake (*Figure 8A–D*). In contrast, stimulation of projections from PSTN^Tac1 neurons to the CeA (*Figure 8E–H*), PVT (*Figure 8I–L*), PBN (*Figure 8M–P*), and NTS (*Figure 8Q–T*) caused decreases in food consumption during the stimulation period. Because it is unknown whether PSTN neurons send collateral projections to multiple brain regions, it is possible that stimulation in a single projection target causes antidromic activation to one or more other target areas. Therefore, these results indicate that PSTN^Tac1 neurons with projections to the CeA, PVT, PBN, and NTS can suppress food intake, although the exact functional role of each downstream target region on food intake behavior remains undetermined.

## Discussion

Taken together, our results show that the PSTN can be subdivided into two nearly distinct subpopulations (*Figure 2*). Both PSTN^Tac1 and PSTN^CRH neurons increase activity in response to food exposure and administration of anorexigenic hormones (*Figures 3 and 4*). Inhibition of PSTN^Tac1 neurons, but not PSTN^CRH neurons, attenuates the full anorexigenic effects of these hormones (*Figure 5*). Additionally, optogenetic or chemogenetic stimulation of PSTN^Tac1 neurons, but not PSTN^CRH neurons, reversibly reduces food consumption (*Figure 6*). Consistently, these two populations have differential expression patterns, with PSTN^Tac1 neurons uniquely projecting to downstream regions that suppress appetite including the PBN, NTS, and PVT (*Figure 7*). Stimulation of projections to these downstream regions also reduces food consumption (*Figure 8*). Taken together, these anatomical and functional results demonstrate that activity in PSTN^Tac1 neurons negatively regulates food consumption. We therefore propose that exposure to food or an increase in anorexigenic hormones causes a transient activation of PSTN^Tac1 neurons, which signals to downstream populations via both glutamatergic and peptidergic signaling to reduce food intake.

We note that our measurements of PSTN neuron activity following refeeding or presentation of food (*Figure 3*) utilized solid mouse chow while experiments that tested the effects of inhibition or

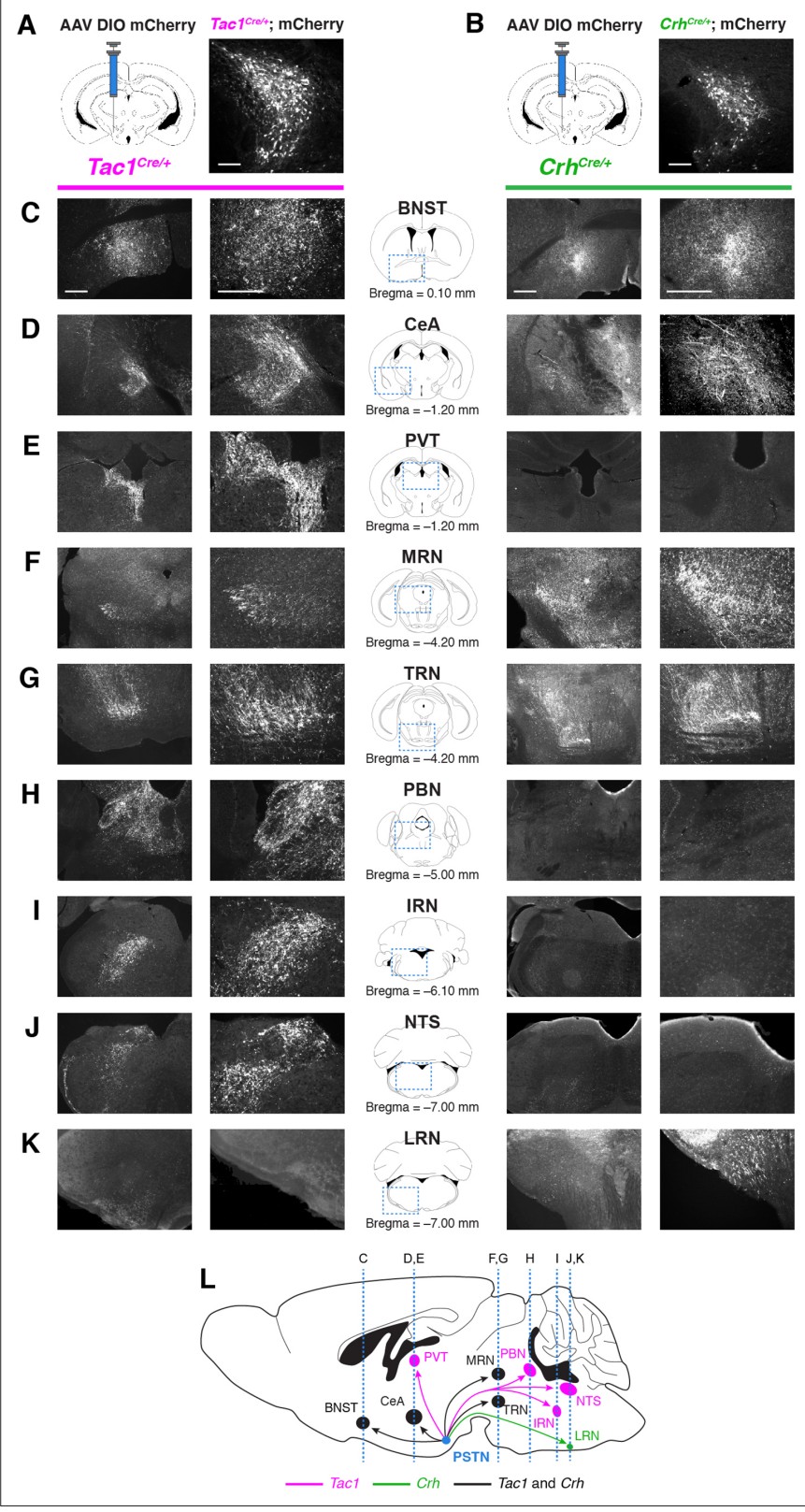

**Figure 7.** Nonidentical efferent projections from PSTN[Tac1] and PSTN[CRH] neurons throughout the brain. (**A–B**) Left, diagram showing viral injection strategy to unilaterally target (**A**) PSTN[Tac1] or (**B**) PSTN[CRH] neurons with the mCherry transgene. Right, representative images of mCherry expression. N=4 mice in each condition. Scale bar, 100 μm. (**C–K**) Representative lower and higher magnification images of mCherry expression in PSTN[Tac1] projections (left

*Figure 7 continued*

columns) and PSTN^CRH projections (right columns) throughout the brain. Scale bars, 500 μm. (**L**) Sagittal mouse brain diagram summarizing locations of PSTN^Tac1 and PSTN^CRH projections throughout the brain. BNST, bed nucleus of the stria terminalis; CeA, central nucleus of the amygdala; IRN, intermediate reticular nucleus; LRN, lateral reticular nucleus; MRN, midbrain reticular nucleus; NTS, nucleus of the solitary tract; PBN, parabrachial nucleus; PVT, paraventricular thalamic nucleus; TRN, tegmental reticular nucleus.

stimulation of PSTN neural activity on food intake behavior (**Figures 5, 6 and 8**) utilized a more palatable liquid diet. We chose a liquid diet to more precisely measure food consumption, but this diet differs in texture and nutritional content with a notably higher carbohydrate content. It is therefore likely that this liquid diet affects stomach distention and hormone release differently from solid chow.

## Effects of stimulation of PSTN neurons

Stimulation of PSTN^Tac1 neurons decreased food intake but did not eliminate consumption altogether (**Figure 6**). These effects are comparable to the reported effects of stimulating hypothalamic POMC neurons, which require several hours of stimulation for a statistically significant decrease in food intake to occur (**Aponte et al., 2011**), or the effects of stimulating PBN^CGRP neurons, which cause an immediate but not absolute reduction in feeding (**Carter et al., 2013**). Because PSTN^Tac1 neurons are glutamatergic (**Figure 2D and E**), they likely decrease feeding by increasing activity in downstream anorexigenic neural populations. In addition to glutamatergic stimulation, tachykinin-1 is alternatively spliced into four neuropeptides including substance P, which has been shown to suppress feeding when centrally administered in rodents and birds (**Dib, 1999**; **Pauliukonis et al., 2020**; **Tachibana et al., 2010**). Because the effects of optogenetic stimulation of PSTN^Tac1 neurons did not decrease food consumption during the post-stimulation period (**Figure 6E**), these neurons may primarily signal via fast glutamatergic neurotransmission, however, the specific contributions of glutamate versus neuropeptide release remain unknown. Future studies should test the effects of PSTN^Tac1 neuron stimulation in Tac1^{-/-} animals or during concurrent administration of specific neuropeptide receptor antagonists.

PSTN^CRH neurons exhibited an increase in neural activity following exposure to food stimuli, but stimulation or inhibition of these neurons caused no noticeable effects on food intake behavior. Therefore, these neurons may be involved in aspects of food intake not studied here, such as regulating nutrient intake or responding to food palatability. Interestingly, Zhu et al. found that deficiency of indispensable amino acids caused substantial upregulation of *Crh* transcript in PSTN^CRH neurons (**Zhu et al., 2012**), suggesting that these neurons may mediate appetite for specific nutrients. Alternatively, these neurons may regulate conditioned associations between specific foods and aversive or appetitive cues.

Previous studies identified downstream projections from PSTN neurons to other anorexigenic neural populations (**Barbier et al., 2020**; **Zhang and van den Pol, 2017**), and our present results distinguish between projections from PSTN^Tac1 and PSTN^CRH neurons. We initially identified the PSTN as a potential anorexigenic population by determining monosynaptic sources of input to PBN^CGRP neurons (**Figure 1**), subsequently determining that these projections originate from PSTN^Tac1 neurons. Consistently, stimulation of these projections reduces food consumption (**Figure 8M–P**). Zhang et al. previously showed that stimulation of glutamatergic projections from the PSTN to the PVT reduces food consumption (**Zhang and van den Pol, 2017**), and our consistent results (**Figure 8I–L**) suggest that these projections originate from PSTN^Tac1 neurons. We also determined that stimulation of projections from PSTN^Tac1 neurons to the CeA (**Figure 8E–H**) and NTS (**Figure 8Q–T**) reduce food consumption. Because we do not yet know whether individual PSTN^Tac1 neurons project to one or more downstream regions, we cannot rule out the potential effects of antidromic activation of each projection target, although stimulation of projections to the BNST caused no observable effects.

## Role of PSTN neurons in mediating anorexigenic hormones

In addition to food stimuli, we found that administration of the anorexigenic hormones amylin, CCK, and PYY caused activation of PSTN^Tac1 and PSTN^CRH neurons using *Fos* expression analysis (**Figure 4A–E**). Likewise, using fiber photometry, we observed a transient increase in activity in PSTN^Tac1 neurons following administration of these hormones via an intraperitoneal catheter (**Figure 4F–L**). Interestingly, chemogenetic inhibition of PSTN^Tac1 neurons attenuated but did not

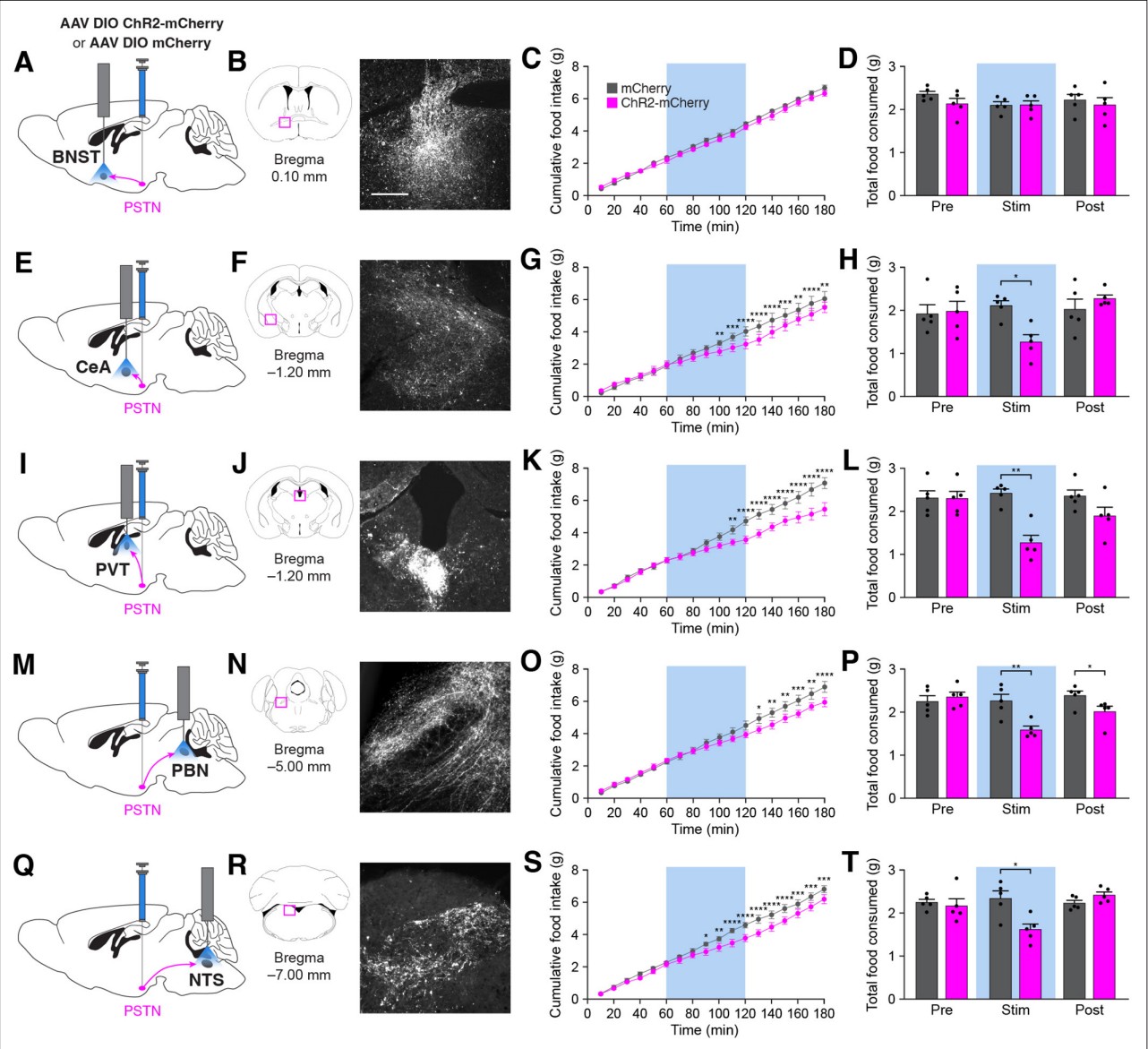

**Figure 8.** Stimulation of PSTN[Tac1] neuron projections suppresses feeding. (**A**) Diagram showing optogenetic strategy to unilaterally target PSTN[Tac1] neurons with ChR2-mCherry or mCherry transgenes with ipsilateral optic fiber implantation above the BNST. (**B**) Representative photomicrograph showing PSTN[Tac1] projections expressing ChR2-mCherry in the BNST. (**C–D**) Quantification of (**C**) cumulative food intake and (**D**) total food consumed during pre-stimulation, stimulation, and post-stimulation of PSTN[Tac1] projections to the BNST. (**E**) Diagram showing optogenetic strategy to unilaterally target PSTN[Tac1] neurons with ChR2-mCherry or mCherry transgenes with ipsilateral optic fiber implantation above the CeA. (**F**) Representative photomicrograph showing PSTN[Tac1] projections expressing ChR2-mCherry in the CeA. (**G–H**) Quantification of (**G**) cumulative food intake and (**H**) total food consumed during pre-stimulation, stimulation, and post-stimulation of PSTN[Tac1] projections to the CeA. (**I**) Diagram showing optogenetic strategy to unilaterally target PSTN[Tac1] neurons with ChR2-mCherry or mCherry transgenes with ipsilateral optic fiber implantation above the PVT. (**J**) Representative photomicrograph showing PSTN[Tac1] projections expressing ChR2-mCherry in the PVT. (**K–L**) Quantification of (**K**) cumulative food intake and (**L**) total food consumed during pre-stimulation, stimulation, and post-stimulation of PSTN[Tac1] projections to the PVT. (**M**) Diagram showing optogenetic strategy to unilaterally target PSTN[Tac1] neurons with ChR2-mCherry or mCherry transgenes with ipsilateral optic fiber implantation above the PBN. (**N**) Representative photomicrograph showing PSTN[Tac1] projections expressing ChR2-mCherry in the PBN. (**O–P**) Quantification of (**O**) cumulative food intake and (**P**) total food consumed during pre-stimulation, stimulation, and post-stimulation of PSTN[Tac1] projections to the PBN. (**Q**) Diagram showing optogenetic strategy to unilaterally target PSTN[Tac1] neurons with ChR2-mCherry or mCherry transgenes with ipsilateral optic fiber implantation above the NTS. (**R**) Representative photomicrograph showing PSTN[Tac1] projections expressing ChR2-mCherry in the NTS. (**S–T**) Quantification of (**S**) cumulative food intake and (**T**) total food consumed during pre-stimulation, stimulation, and post-stimulation of PSTN[Tac1] projections to the NTS. Data represent mean ± standard error. Dots represent individual experimental animals; N=6 mice in all experiments. Blue background represents the 60 min photostimulation period. Post hoc comparisons: *p<0.05, **p<0.01, ***p<0.001, ****p<0.0001. See **Supplementary file 1** for additional statistical information. Scale bar, 250 µm.

*Figure 8 continued on next page*

*Figure 8 continued*

The online version of this article includes the following source data for figure 8:

**Source data 1.** This file contains the raw data for the visualization of data presented in *Figure 8*.

completely rescue the reduction in food intake caused by these hormones (*Figure 5A–J*), indicating that PSTN[Tac1] neuron activity is partially necessary for the appetite-suppressing effects of these hormones. In support of these results, a recent study reported that chemogenetic inhibition of glutamatergic PSTN neurons decreases the appetite-suppressing effects of CCK (*Sanchez et al., 2022*). These results are similar to the effects of inhibiting PBN[CGRP] neurons, which also attenuates but does not completely rescue decreases in food consumption following administration of anorexigenic compounds (*Campos et al., 2016*; *Carter et al., 2013*).

It is interesting that chemogenetic inhibition of PSTN[Tac1] neurons caused an increase in food intake behavior over a sustained period even though the recorded calcium dynamics in these neurons occurred over a relatively short period (*Figure 4G–L*). Perhaps, the neuropeptides released by PSTN[Tac1] neurons cause a sustained change in neural activity in downstream neural populations. Indeed, many other neural populations that affect food intake behavior also show a transient increase in neural activity on the timescale of seconds (*Berrios et al., 2021*; *Luskin et al., 2021*; *Mohammad et al., 2021*; *Wu et al., 2022*). Alternatively, perhaps PSTN[Tac1] neurons exhibit a transient increase in activity but do not completely return to baseline levels over a sustained period in a way that is not detectable by fiber photometry. Indeed, comparisons of fiber photometry versus electrophysiology recordings of other neuronal populations, such as AgRP neurons (*Chen et al., 2015*; *Mandelblat-Cerf et al., 2015*), demonstrate that increased activity of neurons may persist even when fiber photometry recordings return to baseline.

It is also noteworthy that chemogenetic inhibition of PSTN[Tac1] neurons attenuates the effects of amylin, CCK, and PYY by decreasing the frequency of meals as opposed to meal size or meal duration (*Figure 5*). Previous studies of these anorexigenic hormones, especially amylin and CCK, indicate that they affect food intake primarily by decreasing meal size as opposed to meal frequency (*Drazen and Woods, 2003*; *Lutz et al., 1995*; *West et al., 1987*). Therefore, inhibition of PSTN[Tac1] neurons might attenuate the effects of these hormones indirectly, perhaps by reducing activity in downstream populations such as the NTS or PBN. In this model, infusion of anorexigenic hormones activate PSTN[Tac1] neurons that, in turn, cause sustained activation of downstream populations. Without this sustained activity, downstream populations may not have sufficient activity to cause a reduction in the inter-meal interval, leading to increased bouts of feeding. The mechanism by which anorexigenic hormones activate PSTN[Tac1] neurons, as well as how decreases in PSTN[Tac1] neuronal activity affect downstream populations, are important topics for future investigation.

The necessity of the PSTN for the full suppression of appetite following administration of anorexigenic hormones is similar to the findings of *Barbier et al., 2020*, who report that PSTN neurons increase Fos expression in response to novel food exposure, to administration of the anorexigenic compound lipopolysaccharide (LPS), or to the anticancer chemotherapy drug cisplatin. Interestingly, chemogenetic inhibition of PSTN[Tac1] neurons also attenuated but did not completely rescue the reduction in food intake caused by LPS. Inhibition also reduced the reduction of food consumption following exposure to a novel taste. Thus, the PSTN seems to be partially necessary for the suppression of food intake following a wide variety of stimuli including endogenous anorexigenic hormones, exogenous compounds that induce sickness, and behaviors such as taste neophobia. Interestingly, these stimuli all cause activation of neurons in regions downstream of the PSTN including the CeA, NTS, and PBN. However, it is important to note that other studies show that PSTN neurons do not express Fos following administration of LiCl, a compound that causes transient visceral malaise, nor sensory exposure to the bitter tastant quinine hydrochloride (*Yasoshima et al., 2006*). Future studies should examine a diverse range of stimuli that activate these neurons, as well as whether PSTN neuron activity is necessary for their full effects.

# Materials and methods

## Key resources table

| Reagent type (species) or resource | Designation | Source or reference | Identifiers | Additional information |
|---|---|---|---|---|
| Genetic reagent (*Mus musculus*) | B6;129S-Tac1[tm1.1(cre)Hze]/J | The Jackson Laboratory | JAX: 021877 | *Tac1[Cre]* mouse |
| Genetic reagent (*Mus musculus*) | B6(Cg)-Crh[tm1(cre)Zjh]/J | The Jackson Laboratory | JAX: 012704 | *Crh[Cre]* mouse |
| Genetic reagent (*Mus musculus*) | B6.Cg-Calca[tm1.1(cre/EGFP)Rpa]/J | The Jackson Laboratory | JAX: 033168 | *Calca[Cre]* mouse |
| Genetic reagent (Dependoparvovirus) | AAV9-hSyn-DIO-mCherry | Addgene | #50,459 | Adeno-associated viral vector |
| Genetic reagent (Dependoparvovirus) | AAV9-CAG-Flex-GCaMP6s | Addgene | #100,842 | Adeno-associated viral vector |
| Genetic reagent (Dependoparvovirus) | AAV9-hSyn-DIO-hM3Dq-mCherry | Addgene | #44,361 | Adeno-associated viral vector |
| Genetic reagent (Dependoparvovirus) | AAV9-hSyn-DIO-hM4Di-mCherry | Addgene | #44,362 | Adeno-associated viral vector |
| Genetic reagent (Dependoparvovirus) | AAV5-Ef1a-DIO-ChR2-mCherry | UNC North Carolina Vector Core | #AV4379J | Adeno-associated viral vector |
| Genetic reagent (Dependoparvovirus) | AAV5-Ef1a-Flex-TVA-mCherry | Stanford University Gene Vector and Virus Core | #GVVC-AAV-67 | Adeno-associated viral vector |
| Genetic reagent (Dependoparvovirus) | AAV8-CAG-Flex-RabiesG | Stanford University Gene Vector and Virus Core | #GVVC-AAV-59 | Adeno-associated viral vector |
| Genetic reagent (Rabies lyssavirus) | SAD-ΔG-eGFP | Salk Institute Viral Vector Core | SAD-ΔG-eGFP | Modified rabies viral vector |
| Antibody | anti-Fos (Rabbit polyclonal) | Cell Signaling Technology | #2,250 | (1:1000) |
| Antibody | AlexaFluor 488-conjugated anti-rabbit (Donkey polyclonal) | Jackson ImmunoResearch | #711-545-152 | (1:250) |
| Chemical compound, drug | Amylin | Bachem | #H-9475 | 10 µg/kg |
| Chemical compound, drug | Cholecystokinin | Bachem | #4033010 | 10 µg/kg |
| Chemical compound, drug | Protein YY | Bachem | #H-6042 | 100 µg/kg |
| Chemical compound, drug | Clozapine-N-oxide | Sigma-Aldrich | #C0832 | 0.3 mg/kg |
| Chemical compound, drug | 2,2,2-Tribromoethanol | Sigma-Aldrich | #48,402 | |
| Sequence-based reagent | RNAscope Probe-Mm-*Crh* | ACDBio | #316,091 | |
| Sequence-based reagent | RNAscope Probe-Mm-*Tac1* | ACDBio | #410,351 | |
| Sequence-based reagent | RNAscope Probe-Mm-*Tac1*-C3 | ACDBio | #410351-C3 | |
| Sequence-based reagent | RNAscope Probe-Mm-*Slc17a6*-C2 | ACDBio | #319171-C2 | |
| Sequence-based reagent | RNAscope Probe-Mm-*Calb1*-C2 | ACDBio | #428431-C2 | |
| Sequence-based reagent | RNAscope Probe-Mm-*Calb2*-C2 | ACDBio | #313641-C2 | |

*Continued on next page*

*Continued*

| Reagent type (species) or resource | Designation | Source or reference | Identifiers | Additional information |
|---|---|---|---|---|
| Sequence-based reagent | RNAscope Probe-Mm-*Pvalb*-C2 | ACDBio | #421931-C2 | |
| Sequence-based reagent | RNAscope Probe-Mm-*Fos*-C2 | ACDBio | #316921-C2 | (1:10) |
| Commercial assay, kit | RNAscope Multiplex Fluorescent Reagent Kit v2 | ACDBio | #323,100 | |
| Software, algorithm | Fusion Software | Omnitech Electronics | | Food intake measurement software |
| Software, algorithm | Synapse Software | Tucker-Davis Technologies | | Fiber photometry measurement software |
| Software, algorithm | MATLAB | Mathworks | | Fiber photometry measurement data analysis |
| Software, algorithm | Prism 8.0 | GraphPad | | Statistical software |
| Other | Opal 520 | Akoya Biosciences | #FP1487001KT | Histological stain for RNAscope (1:1500) |
| Other | Opal 570 | Akoya Biosciences | #FP1488001KT | Histological stain for RNAscope (1:1500) |
| Other | Opal 690 | Akoya Biosciences | #FP1497001KT | Histological stain for RNAscope (1:1500) |
| Other | DAPI Fluoromount-G | Southern Biotech | #0100–20 | Slide mounting medium with DAPI nuclear stain |
| Other | Fluoromount-G | Southern Biotech | #0100–01 | Slide mounting medium |

## Animals

All experiments were approved by the Institutional Animal Care and Use Committee at Williams College and were performed in accordance with the guidelines described in the U.S. National Institutes of Health *Guide for the Care and Use of Laboratory Animals*. We used *Tac1*$^{Cre/+}$ mice (*Harris et al., 2014*) (Jackson Labs, #021877), *Crh*$^{Cre/+}$ mice (*Taniguchi et al., 2011*) (Jackson Labs, #012704), and *Calca*$^{Cre/+}$ mice (*Carter et al., 2013*) (Jackson Labs, #033168) bred on a C57Bl/6 background. Each experimental group was composed of a randomized selection of mice with identical sex ratios and ages. To comply with NIH guidelines for using both sexes of animals (*Clayton and Collins, 2014*), we used an equal number of male and female animals across data sets. All mice were 7–9 weeks old at the time of surgery and no more than 16–20 weeks old at the cessation of experiments. During experimental procedures, mice were housed in individual cages with a 12 hr/12 hr light/dark cycle at 22 °C.

## Virus preparation

Cre-inducible recombinant adeno-associated virus (AAV) vectors carrying mCherry (AAV9-hSyn-DIO-mCherry, #50459), GCaMP6s (AAV9-CAG-Flex-GCaMP6s, #100842), hM3Dq-mCherry (AAV9-hSyn-DIO-hM3Dq-mCherry, #44361), and hM4Di-mCherry (AAV9-hSyn-DIO-hM4Di-mCherry, #44362) were obtained from Addgene. Cre-inducible AAV carrying ChR2-mCherry (AAV5-Ef1a-DIO-ChR2-mCherry, #AV4379J) was obtained from the Vector Core at the University of North Carolina at Chapel Hill. Cre-inducible AAV carrying TVA-mCherry (AAV5-EF1a-Flex-TVA-mCherry, #GVVC-AAV-67) and rabies glycoprotein (RG, AAV8-CAG-Flex-RabiesG, #GVVC-AAV-59) were obtained from the Neuroscience Gene Vector and Virus Core at Stanford University. Modified rabies virus carrying ΔG-eGFP (SAD-ΔG-eGFP) was obtained from the Viral Vector Core of the Salk Institute. Viral aliquots were stored at –80 °C before stereotaxic injection.

## Stereotaxic surgery

Mice were anesthetized with 4% isoflurane (Henry Schein Animal Health) and placed on a stereotaxic frame (David Kopf Instruments). Once on the frame and throughout the remainder of surgical

procedures, mice received 1–2% isoflurane trans-nasally. The skull was exposed and leveled in the horizontal plane. For viral targeting of transgenes to the PSTN, AAV was stereotaxically injected unilaterally or bilaterally, as described in the text, into the PSTN [anteroposterior (AP), –2.4 mm; mediolateral (ML),±1.1 mm; dorsoventral (DV), –5.25 mm]. A total of 0.5 µl of virus was injected at a rate of 0.1 µl/min and was allowed 8–10 min to diffuse before the injection needle was removed. For modified rabies retrograde labeling in the PBN, AAV Flex TVA-mCherry and AAV DIO RG vectors were unilaterally injected into the PBN (AP, –4.9 mm; ML, 1.4 mm; DV, 3.8 mm); 2 weeks later, SAD ΔG-eGFP was injected into the same location.

Following viral injection, mice used for fiber photometry experiments received unilateral surgical implantation of a mono fiber-optic cannula (Doric Lenses) above the PSTN (AP, –2.4 mm; ML, 1.1 mm; DV, –4.85 mm). Mice used for optogenetic experiments likewise received unilateral surgical implantation of a mono fiber-optic cannula above the PSTN (AP, –2.4 mm; ML, 1.1 mm; DV, –4.85 mm), the NTS (AP, –7.0 mm; ML, 1.0 mm; DV, –3.8 mm), the PBN (AP, –5.2 mm; ML, 1.6 mm; DV, –3.0 mm), the PVT (AP, –1.2 mm; ML, 0.1 mm; DV, –2.6 mm), the CeA (AP, –1.2 mm; ML, 2.4 mm; DV, –4.0 mm), or the BNST (AP, 0.1 mm; ML, 0.8 mm; DV, –4.5 mm). The cannulae were fixed onto the skull with C&B Metabond (Parkell) and dental acrylic.

All mice were allowed at least 14 days to recover from surgery before the start of experimental procedures. Following behavioral experiments, brain sections containing the PSTN or PBN were examined for expression of virus and proper implantation of fiber-optic cannulae. Animals that did not show viral expression (mCherry or eGFP fluorescence) or proper cannulae placement were not included in subsequent data analysis. For anterograde and retrograde labelling experiments, animals were perfused at least two or three weeks after injection, respectively, to allow for maximal viral expression of fluorescent transgenes.

## Intraperitoneal catheter surgery

Two weeks following stereotaxic injection of virus, mice used for fiber photometry experiments following hormone administration received surgical implantation of an intraperitoneal catheter. Catheters were made from hollow polyurethane tubing (0.015 × 1.043 inches; Instech, BTPE-20) and sterilized 24 hr prior to surgery. Mice were anesthetized with 2.5% isoflurane (Henry Schein Animal Health) through a nose cone. Hair was removed from the animal's ventral abdomen and dorsal back near the scapulae using Nair Hair Remover. A small incision was made between the scapulae and the skin was bluntly dissected from the subcutaneous tissue toward the left flank. A transverse abdominal skin incision was made and the skin was bluntly dissected from the subcutaneous tissue toward the flank to complete a subcutaneous tunnel between the two skin incisions. A sterilized catheter attached to a button (Instech, #1-VABM1B/25) was pulled through the tunnel using a hemostat. A small incision was made into the abdominal cavity and the tip of the catheter was placed into the incision site and sutured into place. The button was inserted into the incision made between the scapulae under the skin and was secured with sutures. The absence of leakage was confirmed by injecting 0.9% saline into the catheter and looking for liquid on the outside of the mouse. The abdominal muscle was sutured and the skin incision was closed in two layers. A magnetic cap (Instech, #1-VABM1C) was placed onto the button to prevent foreign materials from entering the intraperitoneal cavity through the button. Animals were allowed another 14 days to recover from surgery before the start of infusion experiments.

## Food intake measurements

For feeding assays, mice were individually housed in specialized food/liquid intake measurement cages attached to water bottles mounted on scales (DietMax, Omnitech Electronics). Mice were provided a liquid diet of Vanilla Ensure (Abbott Laboratories) diluted in a 1:1 ratio with water for a total caloric density of 450 kcal/L. Bottles containing liquid diet were washed and disinfected daily and fully replenished at the beginning of the light cycle. Mice were also provided ad libitum access to HydroGel (ClearH$_2$O, #70-01-5022) to ensure constant hydration. Individual feeding bouts were recorded using scale measurements (Fusion Software, Omnitech Electronics). A meal was defined as any linear decrease greater than 0.02 g of liquid Ensure. The minimum time between two meals was 15 s (otherwise, only one meal was recorded). Mice were allowed to habituate for a minimum of 72 hr

prior to the beginning of experiments, and all food intake measurements were obtained during the middle 4 hr of the inactive cycle.

## Pharmacology

All compounds were prepared in 0.9% sterile saline (VWR, #100216) and stored at –20 °C before use. Compounds consisted of 0.9% sterile saline, amylin (10 µg/kg; Bachem, #H-9475), CCK (10 µg/kg, Bachem, #4033010), or PYY (100 µg/kg; Bachem, #H-6042). For fluorescent in situ hybridization experiments, these compounds were injected intraperitoneally using a 25 G syringe 45 min before anesthesia and perfusion. For chemogenetic experiments, mice received intraperitoneal injection of clozapine-N-oxide (CNO; 0.3 mg/kg, Sigma-Aldrich, #C0832) 10 min prior to injection of an anorexigenic compound followed immediately by food intake recordings. Each mouse was used for a total of 5 experimental sessions, and the mean food intake value for each mouse was included for data analysis across all mice within an experimental group.

## Fiber photometry

All fiber photometry experiments were performed in clear circular chambers with fresh bedding for each trial. Mice implanted with mono fiber-optic cannulae were attached to optical patchcords (400 µm core, 0.48 NA, 1 m long; Doric Lenses) via zirconia connectors (Doric Lenses, Sleeve_ZR_2.5_BK) at least 30 min before each trial to allow for habituation.

During each trial, a GCaMP excitation wavelength of 465 nm blue light modulated at 566 Hz was delivered through the patchcord. A control wavelength of 405 nm violet light modulated at 211 Hz was also delivered to detect calcium-independent GCaMP fluorescence or photobleaching. Delivered frequencies were offset to mitigate contamination or interference from electrical noise in the testing room. Excitation and control lights were generated from light emitting diodes (LEDs; Tucker-Davis Technologies, CLED_465 and CLED_405) and processed through a real-time amplifier (Tucker-Davis Technologies, RZ5P). Fluorescence signals were detected by a visible femtowatt photoreceiver (Tucker-Davis Technologies, Model 2151) with gain set to DC low. The light was then converted to electrical signals and demodulated by a real-time processor (Tucker-Davis Technologies, RZ5P). Data were recorded using Synapse software (Tucker-Davis Technologies).

To test the effects of food exposure on PSTN activity, mice were either fasted for 18 hr or fed ad libitum. After acclimating to the cage, a 30 min baseline recording was produced to ensure a stable signal and decrease photobleaching. At time t=0, mice were provided with either standard laboratory chow pellets, a small scoop of peanut butter, or a novel object (a mini screwdriver or roll of tape). To eliminate any effects of novelty, mice were provided peanut butter in their home cages for at least 2 days prior to testing. In some experiments, mice were water-deprived for 16 hr and at time t=0 a water port was placed in the cage.

To test the effects of peripheral hormone administration, mice were food deprived for 18 hr prior to acclimating in the testing chamber. Amylin, CCK, PYY, or 0.9% saline was delivered by intraperitoneal catheter at a total volume of 300 µl. Following hormone injection, the photometry recording continued for 20 min.

Data were analyzed using custom MATLAB (MathWorks) scripts (available at https://github.com/MattCarter-WilliamsCollege/FiberPhotometryCode.git) (*Carter, 2022*). The 465 and 405 signals were independently downsampled to 1 Hz and normalized to baseline signals to determine ΔF/F, in which ΔF/F = (F – $F_{baseline}$) / $F_{baseline}$, and $F_{baseline}$ is the median of 30 s baseline recording prior to time zero. No isosbestic normalization was introduced. To eliminate movement and bleaching artifacts, recordings with more than 20% change in the 405 nm signal were excluded from analyses. Plots representing mean ± standard error ΔF/F signals for each experiment and heatmaps representing ΔF/F for each trial were generated in MATLAB. Quantitative data analysis was performed in MATLAB and Prism 8.0 (GraphPad). The maximum ΔF/F was defined as the maximum ΔF/F signal intensity from t=0 to t=75 s. The area under the curve was defined as the integral of ΔF/F from t=0 until the time when ΔF/F returned to 0.

## Optogenetic photostimulation

Mice were attached to fiber optic cables (200 µm core, 0.2 NA, 2 m long; Doric Lenses), coated with opaque heat-shrink tubing, via zirconia connectors (Doric Lenses, Sleeve_ZR_2.5) and allowed

to acclimate for at least 3 days prior to experimental sessions. Cables were attached to a 473 nm blue-light laser (LaserGlow) driven by a Master-8 Pulse Stimulator (A.M.P.I.). Light was delivered in 10ms pulses at 20 Hz for 1 s every 4 s over a 1 hr period. During the testing period, food intake was measured for 1 hr before photostimulation, 1 hr during photostimulation, and 1 hr after photostimulation. Each mouse was used for a total of 5 experimental sessions, and the mean food intake value for each mouse was included for data analysis across all mice within an experimental group.

## Perfusions and sectioning

Mice were anesthetized with intraperitoneal injection of 2, 2, 2-Tribromoethanol (Sigma-Aldrich, #48402) dissolved in Tert-amyl alcohol and sterile 0.9% saline. Mice were then transcardially perfused with cold 0.01 M phosphate buffered saline (PBS), pH 7.4, followed by 4% paraformaldehyde in PBS. The brains were extracted, allowed to postfix overnight in 4% paraformaldehyde at 4 °C, and cryo-protected in 30% sucrose dissolved in PBS for an additional 24 hr at 4 °C. Each brain was sectioned coronally at 30 µm on a microtome (Leica Microsystems) and collected in cold PBS. For projection tracing and fluorescent in situ hybridization experiments, the left side of each brain was marked with a pinhole to ensure images were taken from the same hemisphere in each brain. Brain sections were mounted onto SuperFrost Plus glass slides (VWR, #48311–703) and either immediately used for in situ hybridization experiments, immunohistochemistry experiments, or coverslipped with DAPI Fluoro-mount-G (Southern Biotech, #0100–20) and stored in the dark at 4 °C.

## Fluorescent in situ hybridization

Fluorescent in situ hybridization reactions were performed using an RNAscope Multiplex Fluorescent Reagent Kit v2 (ACDBio, #323100) according to the manufacturer's instructions. Brain sections mounted onto SuperFrost Plus glass slides (VWR, #48311–703) were labeled using a combination of target probes for *Crh* (Probe-Mm-Crh, #316091), *Tac1* (Probe-Mm-Tac1, #410351; or Probe-Mm-Tac1-C3, #410351-C3), *Slc17a6* (Probe-Mm-Slc17a6-C2, #319171-C2), *Calb1* (Probe-Mm-Calb1-C2, #428431-C2), *Calb2* (Probe-Mm-Calb2-C2, #313641-C2), *Pvalb* (Probe-Mm-Pvalb-C2, #421931-C2), and *Fos* (Probe-Mm-Fos-C2; #316921-C2). The *Fos* probe was diluted 1:10 to reduce background hybridization. Fluorophore reagent packs (Akoya Biosciences) including Opal 520 (FP1487001KT), Opal 570 (FP1488001KT), and Opal 690 (FP1497001KT) were diluted to a final concentration of 1:1,500 with TSA buffer. Following staining procedures, slides were coverslipped with Fluoromount-G (Southern Biotech, #0100–01) and stored in the dark at 4 °C until microscopy and imaging.

## Immunohistochemistry

Brain sections were washed three times in PBS with 0.2% Triton X-100 (PBST) for 10 min at room temperature. Sections were then incubated in a blocking solution composed of PBST with 3% normal donkey serum (Jackson ImmunoResearch, #017-000-121) for 15 min at room temperature. For primary antibody exposure, sections were incubated in rabbit anti-Fos (1:1000; Cell Signaling Technology, #2250) in blocking solution overnight at 4 °C. After three 5 min washes in blocking solution, sections were incubated in AlexaFluor 488 donkey anti-rabbit (1:250; Jackson ImmunoResearch, #711-545-152) in block solution for 1 hr at room temperature. Finally, sections were washed three times in PBS. Following staining procedures, slides were coverslipped with DAPI Fluoromount-G (Southern Biotech, #0100–20) and stored in the dark at 4 °C until microscopy and imaging.

## Microscopy

For confirmation of virally targeted reporter expression, analysis of anterograde/retrograde projections, and examination of immunofluorescent labeling, slides were examined using an Eclipse 80i epifluorescent microscope (Nikon) and images were captured using a Retiga 2000R digital camera (QImaging). For FISH experiments, slides were examined using a Leica DMi8 confocal microscope. Single channel and overlay pictures were acquired for each section. The resulting images were minimally processed using Photoshop (Adobe) to enhance the brightness and contrast for optimal representation of the data. All digital images were processed in the same way between experimental conditions to avoid artificial manipulation between different datasets.

Analysis of anterograde or retrograde expression was performed on coronal brain sections spanning the entire rostrocaudal distance of the brain. Fluorescently labelled brain regions were identified

using the Allen Mouse Brain Atlas (*Dong, 2008*) and the Paxinos and Franklin Mouse Brain Atlas (*Franklin and Paxinos, 2013*).

Quantification of colocalization of fluorescently-labelled markers in the PSTN was performed on adjacent sections from ~ –2.3 to –2.84 mm from bregma (18 sections per mouse). The Photoshop *count tool* was used to accurately identify and tally individual neurons. Cell counts were corrected for potential double counting using Abercrombie's formula (*Guillery, 2002*).

## Experimental design and statistical analysis

We used a between-subjects experimental design for all experiments. To determine an effective sample size for statistical comparisons, we used an online power and sample size calculator (https://clincalc.com/stats/samplesize.aspx). Assuming a significance level of 0.05, this calculator shows that with at least four mice per group, we had an 80% confidence level of achieving statistical significance between means of 1.1-fold. We excluded an animal from data analysis if flagged by an animal care technician for health reasons during the experimental period or if post hoc histological analysis showed no viral transduction as indicated by an absence of mCherry or GCaMP fluorescence.

Data were analyzed using Prism 8.0 (GraphPad Software). Statistical tests included one-way ANOVA (*Figure 4E*), one-way ANOVA with repeated measures (*Figure 3K and L*; 4 K-L), two-way ANOVA (*Figure 3C*), two-way ANOVA with repeated measures (*Figure 5D–N*; 6D-G; 6K and L; 8C and D; 8G and H; 8K and L; 8O and P; 8S and T), and unpaired two-tailed *t*-test (*Figure 3D*), as described in the text and *Supplementary file 1*. Graphs were exported from Prism 8.0 to Illustrator (Adobe) for preparation of figures.

## Acknowledgements

This research is supported by NIH grant DK105510 from the National Institute of Digestive and Diabetes and Kidney Diseases (NIDDK) and by National Science Foundation grant 1652060 to MEC. We thank R O'Sullivan for performing initial pilot experiments, S Kanoski, E Noble, A Cortella, and S-Y Kim for advice on RNAscope procedures, and J Cone, SC Doret, N Goldstein, A Alhadeff, and N Betley for advice on fiber photometry.

## Additional information

### Funding

| Funder | Grant reference number | Author |
|---|---|---|
| National Institute of Diabetes and Digestive and Kidney Diseases | R15 DK105510 | Matthew E Carter |
| National Science Foundation | IOS 1652060 | Matthew E Carter |

The funders had no role in study design, data collection and interpretation, or the decision to submit the work for publication.

### Author contributions

Jessica H Kim, Grace H Kromm, Olivia K Barnhill, Jacob Sperber, Lauren B Heuer, Sierra Loomis, Matthew C Newman, Kenneth Han, Theresa B Legan, Formal analysis, Investigation, Methodology, Writing – review and editing; Faris F Gulamali, Data curation, Formal analysis, Methodology, Software; Katharine E Jensen, Data curation, Methodology, Software, Writing – review and editing; Samuel C Funderburk, Investigation, Methodology; Michael J Krashes, Conceptualization, Project administration, Supervision, Writing – review and editing; Matthew E Carter, Conceptualization, Data curation, Formal analysis, Funding acquisition, Investigation, Methodology, Project administration, Resources, Supervision, Visualization, Writing – original draft

### Author ORCIDs

Jessica H Kim http://orcid.org/0000-0002-2498-170X

Grace H Kromm http://orcid.org/0000-0003-0847-5468
Olivia K Barnhill http://orcid.org/0000-0003-1597-9701
Faris F Gulamali http://orcid.org/0000-0002-2973-6594
Katharine E Jensen http://orcid.org/0000-0003-1862-8026
Samuel C Funderburk http://orcid.org/0000-0001-5146-2035
Michael J Krashes http://orcid.org/0000-0003-0966-3401
Matthew E Carter http://orcid.org/0000-0003-1802-090X

### Ethics

All experiments in this study were approved by the Institutional Animal Care and Use Committee (IACUC) at Williams College (protocol #CM-A-19). All experiments were performed in strict accordance with the guidelines described in the Guide for the Care and Use of Laboratory Animals of the National Institutes of Health. All surgery was performed under isoflurane anesthesia, and every effort was made to minimize suffering and animal distress.

### Decision letter and Author response

Decision letter https://doi.org/10.7554/eLife.75470.sa1
Author response https://doi.org/10.7554/eLife.75470.sa2

---

## Additional files

### Supplementary files

• Supplementary file 1. This file contains detailed statistical information for all data analyzed throughout this study.

• Transparent reporting form

### Data availability

Source Data files have been provided for Figures 2-6 and 8 (Figures 1 and 7 do not contain quantitative data). These files contain the numerical data used to generate figures and analyze data. Supplementary File 1 contains a complete description of all statistical tests used, methods of multiple comparisons, and critical values for n, p, and degrees of freedom. All MatLab scripts used to analyze fiber photometry data are freely available at https://github.com/MattCarter-WilliamsCollege/Fiber-PhotometryCode.git, (copy archived at swh:1:rev:08e6b78f05cbc7492579753828440a5728b97b35).

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
