## [Editor Report]

This work has identified a previously unrecognized role of the parasubthalamic nucleus in the regulation of feeding behavior. The combination of genetic and pharmacological approaches nicely demonstrates the physiological role of this group of neurons in regulating appetite. These studies will be of interest to the field and more broadly to the readers of *eLife*.

---

## [Decision Letter]

**Decision letter after peer review:**

Thank you for submitting your article "An essential role for a discrete parasubthalamic nucleus subpopulation in appetite suppression" for consideration by *eLife*. Your article has been reviewed by 3 peer reviewers, one of whom is a member of our Board of Reviewing Editor, and the evaluation has been overseen by John Huguenard as the Senior Editor. The following individuals involved in review of your submission have agreed to reveal their identity: Gina Leinninger (Reviewer #2); J. Nicholas Betley (Reviewer #3).

Essential revisions:

As outlined below, several issues need to be clarified to improve the manuscript.

*Reviewer #1 (Recommendations for the authors):*

(1) Intriguingly, the authors pinpointed that PSTMTac1 neurons may play distinguished roles from other anorexigenic neuronal circuits such as the parabrachial nucleus (PBN) and the central nucleus of the amygdala (CeA) as PSTNTac1 neurons may not be involved in aversive food intake behavior such as malaise. However, PSTMTac1 neurons project to the PBN and CeA, and the stimulation of terminal of PSTNTac1 neurons in the PBN and CeA decrease food intake. Although the authors argued that previous studies have shown that pharmacological compounds, which can induce malaise, do not induce Fos expression PSTN, it does not exclude the possibility that PSTNTac1 neuronal activation cause malaise or other aversive behavior. The authors may want to test this possibility by carrying out the simple experiment like examining if PSTNTac1 neuronal activation alters the amount of Kaolin intake.

(2) It is difficult to reconcile the following data in Figure 6. (A) Because the food intake was changed after 30-40 minutes after optogenetic stimulation (panel D), one can assume that PSTNTac1 neurons use neuropeptide instead of fast neurotransmitter as the authors discussed. (B) Interestingly, optogenetics approaches showed (panel E) that the amount of food intake after post-stimulation was comparable between groups, suggesting that fast neurotransmitters are involved. (C) Chemogenetic data showed that immediate decreases in food intake unlike optogenetics (though there were no significant differences in 0-40 min. It is a little bit odd as the error bars are extremely tight), and the effects lasted for 3 hours (panel K). In general understanding, chemogenetic effects are slow as CNO has to be metabolized, and CNO effects can last 30-90 minutes. The authors may want to discuss the data (especially chemogenetics) with the emphasis of differences between optogenetics and chemogenetics results.

(3) I recognized the superiority of liquid food for the precise measurement of food intake. However, the digestion (and other unknown factors) of liquid food may differ from solid food, which may affect hormone release (like insulin, GLP-1, etc) and eventually food intake behavior. I was wondering if the authors have tried solid food or can add discussion regarding the form of food used in the paper.

*Reviewer #2 (Recommendations for the authors):*

I enjoyed this manuscript and think it is excellent in characterizing an understudied brain area with an array of methods. I have some suggestions regarding the text that the authors might consider, as there are some pieces of data that might bear further interpretation or altered presentation to better convey their meaning.

1) The title claims an "essential" role for a PSTN subpopulation in appetite suppression. To my understanding, an essential neuron is one that, if lesioned or inhibited, would prevent any manifestation of appetite suppression. Yet, inhibiting PSTNTac1 neurons at baseline does not impede feeding. Inhibiting the PSTNTac1 neurons does mitigate the ability of anorectic hormones to suppress feeding, but does not completely prevent it. As such, I would suggest removing "essential" from the title to appropriately categorize the role of the neurons.

2) In Figure 3C, the Crh1:cFos quantitation looks elevated and similar to Tac:cFos quantification, but the qualitative images do not show much, if any, Crh1:Fos. Perhaps more representative Crh1:Fos images could be included.

3) In general, green + cyan colocalization is difficult to appreciate. The authors might consider an alternative pseudo-coloring scheme to make the overlap easier to see.

4) Figure 8: the text suggests that the PSTNTac1 and PSTNCrh populations project to some different areas, but the figure doesn't show all of the same brain areas for BOTH populations. Presenting the same areas would be optimally transparent and more impactful to appreciate the differences of the populations.

5) It would be helpful to state the n's in the figure legends.

---

## [Author Response]

Reviewer #1 (Recommendations for the authors):(1) Intriguingly, the authors pinpointed that PSTMTac1 neurons may play distinguished roles from other anorexigenic neuronal circuits such as the parabrachial nucleus (PBN) and the central nucleus of the amygdala (CeA) as PSTNTac1 neurons may not be involved in aversive food intake behavior such as malaise. However, PSTMTac1 neurons project to the PBN and CeA, and the stimulation of terminal of PSTNTac1 neurons in the PBN and CeA decrease food intake. Although the authors argued that previous studies have shown that pharmacological compounds, which can induce malaise, do not induce Fos expression PSTN, it does not exclude the possibility that PSTNTac1 neuronal activation cause malaise or other aversive behavior. The authors may want to test this possibility by carrying out the simple experiment like examining if PSTNTac1 neuronal activation alters the amount of Kaolin intake.

We thank the Reviewer for this suggestion. We are indeed currently performing experiments for a follow-up study to investigate whether PSTN neural activity is aversive. These experiments include studying the effect of PSTN neuron stimulation on Kaolin intake (as suggested), as well as other indices of aversion including conditioned place avoidance and conditioned flavor avoidance. We would prefer to keep these experiments together in a separate study, which we hope is acceptable to the Reviewer.

(2) It is difficult to reconcile the following data in Figure 6. (A) Because the food intake was changed after 30-40 minutes after optogenetic stimulation (panel D), one can assume that PSTNTac1 neurons use neuropeptide instead of fast neurotransmitter as the authors discussed. (B) Interestingly, optogenetics approaches showed (panel E) that the amount of food intake after post-stimulation was comparable between groups, suggesting that fast neurotransmitters are involved. (C) Chemogenetic data showed that immediate decreases in food intake unlike optogenetics (though there were no significant differences in 0-40 min. It is a little bit odd as the error bars are extremely tight), and the effects lasted for 3 hours (panel K). In general understanding, chemogenetic effects are slow as CNO has to be metabolized, and CNO effects can last 30-90 minutes. The authors may want to discuss the data (especially chemogenetics) with the emphasis of differences between optogenetics and chemogenetics results.

The Reviewer brings up excellent points that are worth discussion.

In our hands, CNO takes about 10 min to exert effects, and these effects often last 4-6 hours, much longer than the 3-h experimental duration for our stimulation and inhibition experiments.

We believe that our optogenetic and chemogenetic experiments are consistent with each other because each resulted in significant decreases in cumulative food intake about 50 min after the onset of stimulation. If we look at 5-min intervals starting from the onset of stimulation, we can observe statistically significant decreases in food intake earlier than 50 min, but we chose to report cumulative food intake because this method of visualizing data was less noisy and easy to perceive. For the optogenetic stimulation, all the significant differences between mCherry- and ChR2-mCherry-transduced animals in the post-stimulation period are due to the accumulated effects of stimulation during the stimulation period (as can be observed in panel 6E). The increased effects of chemogenetic stimulation during time 60-180 min are probably due to the fact that chemogenetic stimulation has not ceased and there is no post-stimulation data to report.

The Reviewer brings up an excellent point that the fact that optogenetic stimulation of PSTN^Tac1^ neurons did not cause a statistically significant effect in the post-stimulation period suggests that these neurons may primarily exert effects due to glutamatergic rather than peptidergic signaling in downstream neurons. We added this point to the Discussion (page 13):

“Because the effects of optogenetic stimulation of PSTN^Tac1^ neurons did not decrease food consumption during the post-stimulation period (Figure 6E), these neurons may primarily signal via fast glutamatergic neurotransmission, however, the specific contributions of glutamate versus neuropeptide release remain unknown. Future studies should test the effects of PSTN^Tac1^ neuron stimulation in Tac1^-/-^ animals or during concurrent administration of specific neuropeptide receptor antagonists.”

(3) I recognized the superiority of liquid food for the precise measurement of food intake. However, the digestion (and other unknown factors) of liquid food may differ from solid food, which may affect hormone release (like insulin, GLP-1, etc) and eventually food intake behavior. I was wondering if the authors have tried solid food or can add discussion regarding the form of food used in the paper.

We agree with the Reviewer that this is an important point. We occasionally tried solid food in some experiments—informal observations suggest that stimulation of PSTN^Tac1^ neurons does indeed reduce food intake in mice eating standard lab chow, but we did not perform these experiments with rigor necessary for publication. We agree that the differences between solid and liquid diet are important to mention, therefore we added a paragraph to the Discussion (page 12):

“We note that our measurements of PSTN neuron activity following refeeding or presentation of food (Figure 3) utilized solid mouse chow while experiments that tested the effects of inhibition or stimulation of PSTN neural activity on food intake behavior (Figures 5, 6, and 8) utilized a more palatable liquid diet. We chose a liquid diet to more precisely measure food consumption, but this diet differs in texture and nutritional content with a notably higher carbohydrate content. It is therefore likely that this liquid diet affects stomach distention and hormone release differently from solid chow.”

Reviewer #2 (Recommendations for the authors):I enjoyed this manuscript and think it is excellent in characterizing an understudied brain area with an array of methods. I have some suggestions regarding the text that the authors might consider, as there are some pieces of data that might bear further interpretation or altered presentation to better convey their meaning.1) The title claims an "essential" role for a PSTN subpopulation in appetite suppression. To my understanding, an essential neuron is one that, if lesioned or inhibited, would prevent any manifestation of appetite suppression. Yet, inhibiting PSTNTac1 neurons at baseline does not impede feeding. Inhibiting the PSTNTac1 neurons does mitigate the ability of anorectic hormones to suppress feeding, but does not completely prevent it. As such, I would suggest removing "essential" from the title to appropriately categorize the role of the neurons.

We thank the Reviewer for this very valid suggestion. We have removed the word “essential” from the Title. Our revised title is now: “A discrete parasubthalamic nucleus subpopulation plays a critical role in appetite suppression.” We have also removed the word “essential” from the Abstract.

2) In Figure 3C, the Crh1:cFos quantitation looks elevated and similar to Tac:cFos quantification, but the qualitative images do not show much, if any, Crh1:Fos. Perhaps more representative Crh1:Fos images could be included.

We appreciate this comment very much because it demonstrates that our coloring scheme depicting overlap between Fos (cyan) and Crh (green) is not ideal. The qualitative image does indeed show a high degree of overlap between Fos and Crh, but it is not easy to perceive. Therefore, we have kept the original images but adopted a new color scheme which we believe is much easier to discern. Please see our response to the next comment, below, which addresses the same issue.

3) In general, green + cyan colocalization is difficult to appreciate. The authors might consider an alternative pseudo-coloring scheme to make the overlap easier to see.

Following the comment above, we have changed our coloring scheme to a format that we think is much easier to perceive. We changed the cyan color to a deeper blue that has better contrast with the green color used to label Crh-expressing neurons. We also changed the settings in our “merged” images such that any colocalization between the blue and either magenta or green channels results in a white color. Therefore, the three individual colors stand out from each other and the colocalization is much easier to discern. We are very thankful to the Reviewer for this suggestion as it makes our histological data in Figures 2-4 much more impactful and improves the visualization of our data.

4) Figure 8: the text suggests that the PSTNTac1 and PSTNCrh populations project to some different areas, but the figure doesn't show all of the same brain areas for BOTH populations. Presenting the same areas would be optimally transparent and more impactful to appreciate the differences of the populations.

We thank the Reviewer for this excellent suggestion and have changed the Figure accordingly (We believe the Reviewer was referring to Figure 7 instead of Figure 8). Figure 7 now presents representative images of the same brain areas for both populations of neurons.

5) It would be helpful to state the n's in the figure legends.

We have edited the figure legends to state N values.